**SciPost Physics Community Reports**                                                        **Submission**

LHCHWG-2025-006

# Rare few-body decays of the Standard Model Higgs boson

**David d'Enterria[1],[★] and Van Dung Le[2],[†]**

**1** CERN, EP Department, Geneva, Switzerland
**2** Ho Chi Minh University, Vietnam

★ david.d'enterria@cern.ch ,    † dunglvht@gmail.com

## Abstract

We present a survey of rare and exclusive few-body decays of the standard model (SM) Higgs boson, defined as those into two to four final particles with branching fractions $\mathcal{B} \lesssim 10^{-5}$. Studies of such decays can be exploited to constrain Yukawa couplings of quarks and leptons, probe flavour-changing Higgs decays, estimate backgrounds for exotic Higgs decays into beyond-SM particles, and/or confirm quantum chromodynamics factorization with small nonperturbative corrections. We collect the theoretical $\mathcal{B}$ values for about 70 unobserved Higgs rare decay channels, indicating their current experimental limits, and estimating their expected bounds in p-p collisions at the HL-LHC. Among those, we include 20 new decay channels computed for the first time for ultrarare Higgs boson decays into photons and/or neutrinos, radiative quark-flavour-changing exclusive decays, and radiative decays into leptonium states. This survey can help guide and prioritize upcoming experimental and theoretical studies of unobserved Higgs boson decays.

# 1 Introduction

The discovery of the Higgs boson at the CERN Large Hadron Collider (LHC) in 2012 [1, 2] constituted a major milestone in establishing the Standard Model (SM) of particle physics. However, despite its remarkable phenomenological success, the SM remains incomplete as it fails to account for outstanding empirical observations —such as matter-antimatter asymmetry, dark matter, and nonzero neutrino masses— that require physics beyond it (BSM) for their explanation [3]. In addition, the SM offers no justification for the large unnatural difference between the Higgs and Planck masses (hierarchy problem), nor it provides insight into the origin, number, and structure (masses and mixings) of quark and lepton flavour families. In this context, scrutinizing the properties of the scalar boson, in particular testing its couplings to all SM (and potential BSM) particles, is a top priority for the physics program of the LHC [4] as well as of future colliders [5].

The proton-proton (p-p) high-luminosity LHC phase (HL-LHC) will achieve large center-of-mass energies (up to $\sqrt{s} = 14$ TeV) and integrated luminosities (up to $\mathcal{L}_{\text{int}} = 2 \times 3$ ab$^{-1}$ for ATLAS + CMS combined) allowing the production of a very large number of Higgs bosons [6]: $N_{\text{HL-LHC}}(\text{H}) = \sigma(\text{pp} \rightarrow \text{H} + \text{X}) \times \mathcal{L}_{\text{int}} \approx 3.5 \cdot 10^8$ bosons, for $\sigma(\text{pp} \rightarrow \text{H} + \text{X}) \approx 60$ pb. Such large data samples will make it possible to measure, or place relevant constraints on, many of its rarest decays. This paper focuses on Higgs decays into two to four final-state particles, with branching fractions below $\mathcal{B} \approx 10^{-5}$, that are not usually included in the standard Higgs decays codes [7] and remain experimentally unobserved to date. Broadly speaking, we consider the three types of few-body decays shown in Fig. 1:

**(i)** decays into three (or four) lighter gauge bosons, or into one (or two) gauge boson plus two neutrinos,

**(ii)** decays into a lighter gauge boson plus a single hadronic (or leptonic) bound system in the form of a meson (or leptonium) state, and

**(iii)** exclusive decays into two onium states.

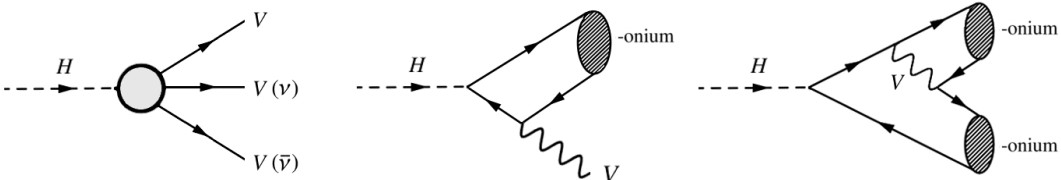

Figure 1: Schematic diagrams of rare and exclusive two- and three-body decays of the Higgs boson into (i) two or three gauge bosons (V = Z, W, $\gamma$) or into a gauge boson plus a neutrino pair ($\nu\bar{\nu}$) through virtual loops (grey circle) (ii) a gauge boson plus a difermion bound state (meson or leptonium), and (iii) two onium states.

The rarity of the first type of decays is due to their proceeding through suppressed heavy-particle loops (potentially sensitive to BSM physics), whereas the second and third decays occur very scarcely because of the smallness of the Yukawa couplings of the Higgs boson to light fermions, and because the probability to form a single (let alone double) -onium bound state, out of the outgoing quarks or leptons of the primary decay, is very suppressed.

There are multiple varied scientific reasons for the study of such decays. A first generic motivation is the possibility that new physics phenomena alter some of these rare partial decay widths. Precision tests of suppressed or forbidden processes in the SM —such as flavour changing neutral currents (FCNC), or processes violating lepton flavour (LFV) or lepton flavour universality (LFUV)— are powerful probes of BSM physics that have been mostly studied in

b-quark decays so far [8,9]. Unlike in the latter case where, due to the relatively low masses of the B hadrons involved, large power corrections lead to sizable theoretical uncertainties to the decay rates, power corrections in Higgs boson decays are under better theoretical control thanks to the large boost of the final-state particles. Secondly, decays into photons and/or neutrinos (Fig. 1, left) lead to experimental signatures that are identical to exotic BSM Higgs decays [4,10], and therefore their SM rates need to be properly estimated as potential backgrounds in new physics searches. Thirdly, exclusive Higgs decays such as those shown in Fig. 1 (center) are sensitive to the Yukawa couplings of the charm and lighter quarks via $H \to q\bar{q}$ [11–15], or of leptons via $H \to \ell^+\ell^-$ [16], as well as to FCNC via $H \to q\bar{q}'$ decays. All such SM decays are very difficult to probe experimentally due to the smallness of the fermion masses involved (and/or their heavily suppressed loop-induced FCNC rates) and the complications associated with quark-flavour identification in inclusive dijet decays, but can be potentially better isolated in exclusive few-body final states. Last but not least, rare decays where the onium state (center and right panels of Fig. 1) decays into diphotons or dileptons, constitute also a background for different searches for exotic BSM decays [17], such as e.g., $H \to a(\gamma\gamma)a(\gamma\gamma)$, or $H \to a(\gamma\gamma)Z$, where $a$ is an axion-like particle (ALP) [18,19] or a massive graviton [20] decaying into photons; or $H \to A'(\ell^+\ell^-)+X$ where a dark photon $A'$ further decays into a lepton pair [21].

Theoretically, the calculation of the partial widths of the few-body decays schematically shown in Fig. 1 (left) is carried out through an expansion of the underlying virtual loops in the electroweak (EW) and/or quantum chromodynamics (QCD) couplings ($\alpha$ and $\alpha_s$, respectively), at a given order of perturbative accuracy. First calculations were performed at leading order (LO), but more recent results exist at next-to-leading-order (NLO) accuracy. With regards to the center and right diagrams of Fig. 1, the formalism of QCD factorization is a well-established approach to calculate rates for hard exclusive processes with individual hadrons in the final state [22–24]. Such a framework separates the process into short-distance partonic interactions calculable within perturbative QCD, and nonperturbative hadron formation described by objects such as light-cone distribution amplitudes (LCDAs, for light hadrons) [25,26], or long-distance matrix elements (LDMEs, for charm and bottom hadrons) in non-relativistic QCD (NRQCD) approaches [27]. The decay amplitude is then obtained through a convolution of the perturbative hard-scattering functions and the nonperturbative LCDAs or LDMEs, extended if needed to include the resummation of large logarithms within soft-collinear effective theory (SCET) [28,29]. Such a formalism allows expansions in the ratio of the hadronization scale to the hard scale, which is small in processes involving the Higgs boson, ensuring that nonperturbative corrections are suppressed, $\mathcal{O}(\Lambda_{QCD}/m_H) \approx 10^{-3}$. Studies of exclusive decays of the H boson not only thus provide a sensitive test of the SM Higgs sector, but also stringent tests of the QCD factorization formalism, including constraints on poorly known aspects of the nonperturbative formation of hadronic bound states.

The first purpose of this work is to present a comprehensive summary of the current status of theoretical and experimental studies of rare and exclusive few-body decays of the SM Higgs boson, based on the estimates presented in our review [16], extended here with more details and new results. We have collected all theoretical calculations and experimental limits for the rates of rare and exclusive decays existing in the literature, revised them, and complemented them with ~20 additional channels estimated for the first time. In total, we provide a list of about 70 predicted rare decays branching ratios, and identify those that are potentially observable at the LHC and those with negligible rates unless some BSM physics enhances them. As of today (mid 2025), the LHC measurements have been able to set limits at 95% confidence level (CL) for about 20 of the ~70 decay branching fractions considered here. These limits are listed in the tables below, including in some cases new results not yet available in the 2024 PDG decays listings [30]. The second key goal of our work is to provide expecta-

tions for the achievable bounds (or potential observation) at the HL-LHC. The corresponding extrapolations are obtained through two different methods:

1. For those decay channels where dedicated ATLAS and/or CMS studies exist that have determined the expected limits at the HL-LHC phase [31–33], we quote directly those (further improved by a factor of $\sqrt{2}$ to account for the statistical combination of two experiments, ATLAS + CMS).

2. For those channels where LHC limits exist today, based on a given integrated luminosity (labelled "$\mathcal{L}_{\text{int}}(13\,\text{TeV})$" below), but for which no dedicated HL-LHC extrapolations exist, we estimate the latter by assuming that they will be statistically improved by the size of the final data sample, namely by the ratio of squared-root integrated luminosities $\sqrt{2 \times 3\,\text{ab}^{-1}}/\mathcal{L}_{\text{int}}(13\,\text{TeV})$, where the factor of two assumes an ATLAS + CMS combination. For individual LHC measurements carried out with the full Run-2 integrated luminosity of $\sim$140 fb$^{-1}$, this translates into an expected HL-LHC improvement of more than a factor of six. Bounds estimated this way are conservative, as they ignore improvements in data analyses (e.g., optimized event selection, extended categorization based on the production mode, adoption of more advanced statistical profiling methods, etc.), increased particle production cross sections (from collision energies rising from 13 to 14 TeV), and possible multi-experiment combinations (e.g., by adding results from LHCb). For such a reason, we also plot indicatively in the figures below the lowest branching fraction theoretically producible, $\mathcal{B} = 1/N_{\text{HL-LHC}}(\text{H}) \approx 3 \cdot 10^{-9}$, at the HL-LHC.

The aim of this document is to motivate and help guide and prioritize upcoming experimental and theoretical studies of multiple rare decay channels of the scalar boson. Our previous review [16] discussed the perspectives at future facilities (mostly FCC-ee and FCC-hh), whereas the present work focuses on those at the (HL-)LHC. The paper is organized as follows. Section 2 presents the results for rare two-, three-, and four-body decays. Section 3 reviews exclusive Higgs boson decays into a gauge boson plus a meson, $\text{H} \to \text{V} + \text{M}$ with $\text{V} = \gamma, \text{Z}, \text{W}^{\pm}$. The Higgs radiative decays into leptonium states, $\text{H} \to (\ell^+\ell^-) + \text{V}$ (with $\text{V} = \gamma, \text{Z}$) are discussed in Section 4, and the decays into two mesons, $\text{H} \to \text{M} + \text{M}$, are reviewed in Section 5. The paper is closed with a summary in Section 6.

## 2   Rare two-, three-, and four-body Higgs boson decays

Figure 2 shows representative diagrams for rare Higgs boson decays into two neutrinos, a photon plus two neutrinos, three or four photons, and a Z boson plus two gluons or two photons. Calculations of rates for various of these processes, which proceed via virtual loops in the SM, do not exist to our knowledge, or were estimated long ago with outdated SM parameters. Since they are very suppressed and have no obvious phenomenological impact, they are not included in the standard Higgs decay codes [7], although they are intriguing for diverse reasons exposed below. We have computed them with the MADGRAPH5_AMC@NLO program [34] (called MG5_AMC, hereafter) with QCD and EW loop corrections up to NLO accuracy [35, 36], including massive fermions in the loops (where needed), using the input parameters discussed in Ref. [16]. The corresponding results are listed in Table 1 and plotted in Fig. 3 as a function of the Higgs boson mass $m_{\text{H}}$ (left) and, for the SM Higgs mass, in negative log scale of the branching fraction (right).

Our first result is that of the invisible two-body Higgs boson decay into a neutrino pair $\text{H} \to \nu\overline{\nu}$, which is infinitesimal in the SM ($\mathcal{B} \approx 10^{-36}$) compared to the standard invisible (four-neutrino) $\text{H} \to \text{ZZ}^{\star} \to 4\nu$ decay ($\mathcal{B} \approx 0.1\%$) [7]. However, since the SM assumption of massless $\nu$'s is invalid, the $\text{H} \to \nu\overline{\nu}$ decay can receive extra contributions depending on

Figure 2: Representative diagrams of rare 2-, 3-, and 4-body decays of the H boson into photons and/or neutrinos, and into Z bosons plus gluons or photons.

the mechanism of neutrino mass generation actually realized in nature. Therefore, it is a process deserving attention in the context of measurements of invisible Higgs decays and/or SM extensions including massive neutrinos. The second diagram of Fig. 2 shows the photon-plus-neutrinos decay, which proceeds through Z and W loops and, since the neutrino pair goes undetected, it appears experimentally as an unbalanced monophoton Higgs decay. Such a final state is shared by many exotic BSM Higgs final states [10], and it is important to determine its SM rate. Its branching fraction is $\mathcal{B} = 3.7 \cdot 10^{-4}$, being the least rare decay considered in this whole survey. Such a decay rate is about 20% larger than the naive estimate given by $\mathcal{B}(H \rightarrow Z\gamma) = 1.54 \cdot 10^{-3}$ [7] multiplied by $\mathcal{B}(Z \rightarrow \nu\overline{\nu}) = 0.20$ [30] because extra W-induced channels (not shown in Fig. 2) contribute to the amplitude.

Table 1: Rare Higgs decays to two neutrinos, a photon plus two neutrinos, three or four photons, and a Z boson plus two $\gamma$'s or two gluons. For each decay, we provide the theoretical branching fraction computed with MG5_AMC. No current (or extrapolated, future) experimental limits exist.

| Decay | | Theoretical branching fraction | Experimental limits current | HL-LHC |
|---|---|---|---|---|
| H → | $\nu + \overline{\nu}$ | $7.2 \ \times 10^{-36}$ (this work) | – | – |
| | $\gamma + \nu + \overline{\nu}$ | $(3.4\text{–}3.7) \ \times 10^{-4}$ (this work), [37] | – | – |
| | $\gamma + \gamma + \gamma$ | $1.0 \ \times 10^{-40}$ (this work) | – | – |
| | $\gamma + \gamma + \gamma + \gamma$ | $5.4 \ \times 10^{-12}$ (this work) | – | – |
| | $Z + \gamma + \gamma$ | $(2.0\text{–}2.4) \ \times 10^{-9}$ (this work), [38, 39] | – | – |
| | $Z + g + g$ | $(3.6\text{–}6.3) \ \times 10^{-7}$ (this work), [38, 39] | – | – |

The third ultrarare Higgs decay considered is $H \rightarrow 3\gamma$. At variance with the naive assumption that a scalar particle cannot decay into three photons, because such a process would violate charge-conjugation (C) symmetry, $C(H) = +1 \neq C(3\gamma) = (-1)^3 = -1$, such a decay is possible, albeit extremely suppressed, for EW-induced decays. In a situation akin to the case of the triphoton decay of para-positronium [40, 41] or of the $\pi^0$ meson [42], which is forbidden in QED but not in the EW theory, such a Higgs decay can proceed through the W box shown in the top-right panel of Fig. 2. Since $H \rightarrow 3\gamma$ violates C-symmetry, it must also violate parity in order to conserve CP and, therefore, the final state must be composed of three spatially symmetric photons with vanishing total angular momentum. As a consequence, the partial width of this channel features an utterly small $\mathcal{O}(10^{-40})$ probability. The box diagram is, however,

not the only theoretical contribution to this decay. It can also proceed via $H \to Z(\gamma\gamma)\gamma$, where the $Z \to \gamma\gamma$ (intermediate) decay "evades" the Landau–Yang selection rule, which forbids the decay of a massive gauge boson into two real massless bosons [43, 44], because the actual physical final state is composed of three, not two, photons. Our MG5_AMC-based estimates indicate that this latter contribution would also be tiny, of the same size as the box one.

Two other rare Higgs boson decays into three gauge bosons are possible: $H \to Zgg$ and $H \to Z\gamma\gamma$. They have been considered before in the literature, albeit in the $m_t \to \infty$ limit and/or with outdated SM parameters [38, 39, 45]. Both decays proceed through a fermion box (mostly, a top quark box) as shown in the bottom of Fig. 2 (left and center diagrams). The $H \to Z\gamma\gamma$ decay rate is much more suppressed than the naive expectation of it being similar to $\mathcal{B}(H \to Z\gamma) \approx 1.5 \cdot 10^{-3}$ times a factor $\alpha \approx 10^{-2}$ for the extra photon emission, because there are no W bosons in the virtual box due to the charge conjugation properties of the gauge boson couplings. The branching fractions obtained for both processes with MG5_AMC (including top-, bottom-quarks, and tau leptons in the box) are $\mathcal{B}(H \to Zgg) = 6.3 \cdot 10^{-7}$ and $\mathcal{B}(H \to Z\gamma\gamma) = 2.4 \cdot 10^{-9}$, respectively. With such low $\mathcal{B}$ values, one expects about 200 $H \to Zgg$ events, and short of one $H \to Z\gamma\gamma$ event, at the HL-LHC. Given the additional event loss when searching for the $\mathcal{B}(Z \to \ell^+\ell^-) \approx 3\%$ decay, both very rare channels are only realistically accessible at a future machine such as FCC-hh [46], where they offer a relatively clean signal of two energetic gluons or photons plus a back-to-back $Z \to \ell^+\ell^-$ pair. It is worth noting that the box contribution to the $H \to \gamma gg$ decay, identical to the $H \to Zgg$ one but exchanging the Z boson by a photon, is forbidden by the generalized Furry's theorem [47] that applies for the photon case but is evaded by the axial coupling of the Z boson to fermions in the $H \to Zgg$ case. The $H \to \gamma gg$ decay can nonetheless proceed through $H \to \gamma Z(gg)$ with an approximate rate of $\mathcal{O}(10^{-40})$, according to estimates obtained with MG5_AMC using the SMEFT@NLO model [48]. This is a negligible rate but not exactly zero because, again, the Landau–Yang theorem (which would seemingly forbid the "intermediate" $Z \to gg$ decay) does not exactly apply for the physical 3-body final-state actually realized in the decay.

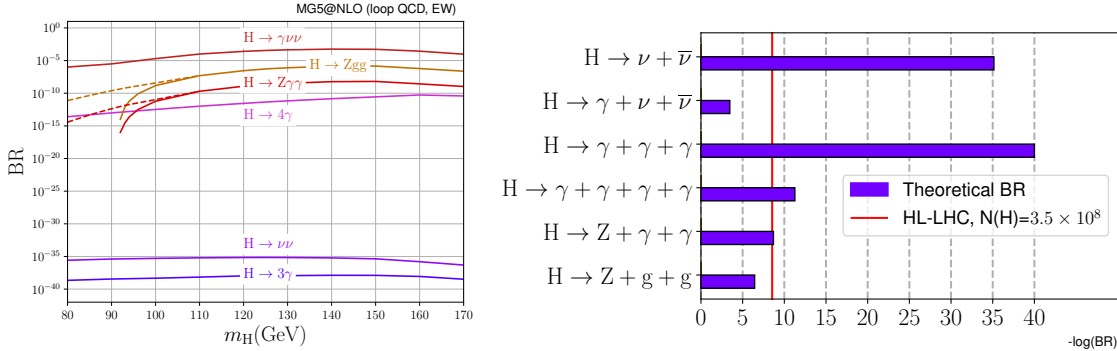

Figure 3: Theoretical branching fractions of the Higgs boson into rare two-, three-, or four- gauge bosons and/or neutrinos shown as a function of the Higgs boson mass (left) and as blue bars in negative log scale (right). In the left panel, the dashed lines for $m_H < m_Z$ show the $H \to Z^*gg$, $Z^*\gamma\gamma$ decays with offshell Z bosons. In the right panel, the red vertical line indicates the minimum $\mathcal{B}$ value reachable at the HL-LHC given just by the total number of H bosons expected to be produced.

Finally, it is interesting to compute the 4-photon decay of the Higgs boson as it may constitute a background for exotic decays into a pair of lighter even-spin particles, each of which further decays into two photons, $H \to a(\gamma\gamma)a(\gamma\gamma)$. The SM decay has a branching fraction of $\mathcal{B}(H \to 4\gamma) = 4.56 \cdot 10^{-12}$, which is 28 orders-of-magnitude larger than the 3-$\gamma$ decay, as C and P conservation is not an issue here, and the rate is "only" suppressed by the presence of

heavy charged-particle loops.

From the experimental perspective, among all rare channels discussed in this section, LHC searches have been performed to date only for the $4\gamma$ final state [49–52] (searches for $3\gamma$ decays have focused on Z' resonances off the Higgs peak [49]), although limits have been set for the H $\rightarrow$ a($\gamma\gamma$)a($\gamma\gamma$) process with two intermediate ALPs decaying into photons. It would be interesting if ATLAS/CMS could recast these searches into bounds on the H $\rightarrow$ 4$\gamma$ "continuum" decay.

# 3   Exclusive Higgs decays into a gauge boson plus a meson

Probing the Yukawa couplings to first-generation (q = u, d) and second-generation (q = s, c) quarks, let alone flavour-changing Higgs decays H $\rightarrow$ q$\bar{q}'$, is very difficult at the LHC due to the smallness of the H $\rightarrow$ q$\bar{q}$, c$\bar{c}$, q$\bar{q}'$ decay widths, the limitations of jet quark-flavour identification, and the enormous QCD dijet backgrounds. As an alternative [11–15], it has then been proposed to constrain those couplings via rare exclusive decays into a gauge boson (V = Z, W$^{\pm}$, $\gamma$) plus a meson, H $\rightarrow$ V + M (Fig. 4), where potential enhanced contributions from new physics effects can be more easily detected as the smaller SM backgrounds make deviations more apparent.

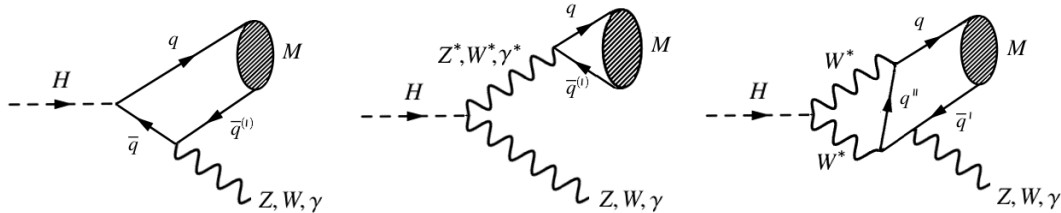

Figure 4: Schematic diagrams of exclusive decays of the H boson into a meson plus a gauge boson in direct (left), indirect (center), and W-loop (right) processes. The solid fermion lines represent quarks, and the gray blob represents the mesonic bound state.

The leftmost diagram of Fig. 4 shows the process where the Higgs boson directly couples to a quark-antiquark pair that radiates a gauge boson and forms the mesonic bound state. The second diagram depicts indirect contributions to the decay amplitude, where the scalar boson decays first into two gauge bosons, one of which transforms (offshell) into a meson via V$^*$ $\rightarrow$ q$\bar{q}$. The rightmost diagram shows the radiative FCNC decay into a gauge boson plus a flavoured meson, through a double W loop. The indirect diagram provides the dominant contribution to the decay rates due to the smallness of the Yukawa couplings to the first- and second-generation quarks, and the sensitivity to the Higgs-quark coupling comes from the (destructive) interference between the two amplitudes. Thus, for example, the H $\rightarrow$ $\gamma$ + J/$\psi$, $\gamma$ + $\phi$ modes allow direct access to the flavour-diagonal coupling of the Higgs boson to the charm and strange quarks, respectively, while the H $\rightarrow$ $\gamma$ + $\rho$, $\gamma$ + $\omega$ decays can probe the Higgs couplings to up and down quarks. In addition, the radiative decays H $\rightarrow$ $\gamma$ + M with M = K$^{*0}$, D$^{*0}$, B$_s^{*0}$, B$_d^{*0}$ provide possibilities to probe flavour-violating q-q' Higgs couplings [13, 53], which in the SM can only proceed through a virtual W boson (Fig. 4, right) or through the direct process (via contact interaction) in BSM scenarios.

The direct amplitudes are much smaller than the indirect ones, except for decays involving the b-quark such as H $\rightarrow$ $\gamma$ + $\Upsilon$. Since the meson mass is $m_M \ll m_H$, the meson is emitted at very high energy $E_M \gg m_M$ in the Higgs boson rest frame. The constituent partons of the meson can thus be described by energetic particles moving collinear to the direction of M, and

QCD factorization, either in the SCET or NRQCD incarnation, can be employed to compute them. The indirect amplitude where the virtual gauge boson transforms into a meson through the matrix element of a local current, $V^* \to M$, can be parameterized in terms of a single parameter: the meson decay constant $f_M$. For vector mesons (VMs), this quantity can be directly obtained from the experimental measurements of their leptonic partial decay widths (which proceed through the EW annihilation $q\bar{q} \to \gamma^*, Z^* \to \ell^+ \ell^-$), given by

$$\Gamma(M \to \ell^+ \ell^-) = \frac{4\pi Q_q^2 f_M^2}{3 m_M} \alpha^2(m_M),$$

where $Q_q$ is the electric charge of the constituent quarks. Corrections due to the offshellness of the photon and to the contribution of the $H \to \gamma Z^*$ process are suppressed by $m_M^2/m_H^2$ and $m_M^2/(m_Z^2 - m_M^2)$, respectively, and hence very small [14].

The branching fraction of a generic $H \to V + M$ decay is obtained from the sum of the two interfering amplitudes, $\mathcal{A}_{dir}$ and $\mathcal{A}_{ind}$,

$$\mathcal{B}(H \to V + M) = |\mathcal{A}_{dir} + \mathcal{A}_{ind}|^2.$$

In $H \to \gamma + M$ decays, the final meson can only be a vector state, but in exclusive decays involving the Z or W bosons, pseudoscalar (PS) and vector (VM) mesons can be produced thanks to the different heavy gauge boson polarizations [54]. These two decay rates can be decomposed as:

$$\mathcal{B}(H \to Z, W + PS) = |\mathcal{A}_{dir} + \mathcal{A}_{ind}|^2 \approx |\mathcal{A}_{ind}|^2 \left[ 1 + 2 \frac{\mathcal{R}(\mathcal{A}_{dir} \mathcal{A}_{ind})}{|\mathcal{A}_{ind}|^2} \right]$$
$$= \mathcal{B}(H \to Z, W + PS)_{ind}(1 + \delta_{dir}). \tag{3}$$

$$\mathcal{B}(H \to Z, W + VM) = |\mathcal{A}^\perp|^2 + |\mathcal{A}^\parallel|^2 = |\mathcal{A}_{dir}^\perp + \mathcal{A}_{ind}^\perp|^2 + |\mathcal{A}_{ind}^\parallel|^2$$
$$\approx \left( |\mathcal{A}_{ind}^\perp|^2 + |\mathcal{A}_{ind}^\parallel|^2 \right) \left[ 1 + \frac{2\mathcal{R}(\mathcal{A}_{ind}^\perp \mathcal{A}_{dir}^\perp)}{|\mathcal{A}_{ind}^\perp|^2 + |\mathcal{A}_{ind}^\parallel|^2} \right]$$
$$= \mathcal{B}(H \to Z, W + VM)_{ind}(1 + \delta_{dir}), \tag{4}$$

where $A^\perp, A^\parallel$ are the amplitudes of the transverse and longitudinal polarization of the VM, respectively, and we have truncated the second $|\mathcal{A}_{dir}|^2$ term because the direct contribution is much smaller than the indirect one. For the longitudinal amplitude decomposition, the direct contribution has been neglected because it is highly suppressed. The final $H \to Z, W + M$ decay expressions can then be written in a simplified form in terms of a small direct correction $\delta_{dir}$ to the indirect branching fraction.

We compare below the decay branching fractions of "exclusive radiative direct" decays versus "inclusive diquark" decays $\mathcal{B}(H \to q\bar{q})$ obtained from the HDECAY code [7]), which are both proportional to the Higgs-quark Yukawa coupling squared, $g_{Hq}^2$. Using the values listed in the next subsections, we anticipate the following ratios for the different exclusive meson types (here below, 'heavy M' ('light M') indicates mesons with (without) heavy quarks):

$$\mathcal{B}(H \to \gamma + M)_{dir} \approx \mathcal{B}(H \to q\bar{q}) \times \begin{cases} 10^{-7} & \text{(heavy M)} \\ 10^{-12} & \text{(light M)} \end{cases} \tag{5}$$

$$\mathcal{B}(H \to Z + PS)_{dir} \approx \mathcal{B}(H \to q\bar{q}) \times \begin{cases} 10^{-12} & \text{(heavy PS)} \\ 10^{-14} & \text{(light PS)} \end{cases} \tag{6}$$

$$\mathcal{B}(H \to Z + VM)_{dir} \approx \mathcal{B}(H \to q\bar{q}) \times \begin{cases} 10^{-8} & \text{(heavy VM)} \\ 10^{-10} & \text{(light VM)} \end{cases} \tag{7}$$

Namely, the exclusive rates are expected to be orders-of-magnitude more suppressed than the inclusive ones because, roughly speaking, one needs to pay a minimum $\alpha \approx 10^{-2}$ penalty to emit an EW boson plus a $\mathcal{O}(10^{-5})$ factor to form a single $(q\bar{q})$ bound state. Nonetheless, the inclusive Higgs diquark decays (in particular for light-quark jets) are virtually impossible to identify on top of the QCD dijet continuum background at the LHC and, therefore, the exclusive decays are still useful in searches although, unless the light-quark Yukawa couplings are enhanced by BSM physics, the direct contributions will be often swamped by indirect ones, as we also quantify below case-by-case.

## 3.1 Higgs decays into a photon plus a neutral vector meson

Table 2 lists the theoretical predictions and experimental limits for Higgs decay $\mathcal{B}$'s into a photon plus a vector meson, and Fig. 5 displays them in graphical form. Theoretical $\mathcal{B}$ values are in the range of $\mathcal{O}(10^{-5}\text{–}10^{-9})$, with larger rates for decays into light-quark and charm vector mesons due to significant indirect contributions, while the radiative bottomonium decays are heavily suppressed due to strong cancellation of direct-indirect contributions (as indicated by the $\mathcal{A}_{\mathrm{dir}}/\mathcal{A}_{\mathrm{ind}}$ amplitude ratios of order one, listed in the table). Experimental bounds have been set for all decays [55–58]. Conservatively, the HL-LHC will be able to set bounds about $3\text{–}10^4$ times above their expected SM branching fractions. With analyses improvements, evidence should be possible at the HL-LHC for a couple of decays, such as $H \to \gamma + \rho$ and $H \to \gamma + J/\psi$, whereas the heaviest bottomonium radiative decays will require the number of Higgs bosons produced at a future machine such as FCC-hh.

Table 2: Exclusive Higgs decay rates into a photon plus a vector meson. For each decay, we provide the theoretical branching fraction(s), the ratios of direct-over-indirect and of exclusive-direct-over-inclusive decay amplitudes, the current experimental limit and that conservatively estimated for HL-LHC (as well as the theory over HL-LHC-expected ratio).

| | Theoretical | | | | Experimental limits | | |
| $H \to \gamma + M$ | branching fraction | | $-\mathcal{A}_{\mathrm{dir}}/\mathcal{A}_{\mathrm{ind}}$ | $\lvert\mathcal{A}_{\mathrm{dir}}/\mathcal{A}_{H\to q\bar{q}}\rvert$ | current | HL-LHC | $\frac{\mathcal{B}(\mathrm{th})}{\mathcal{B}(\mathrm{exp,\,HL\text{-}LHC})}$ |
|---|---|---|---|---|---|---|---|
| $H \to \gamma + \rho^0$ | $(1.68 \pm 0.08) \times 10^{-5}$ | [14] | $10^{-5}$ | $3 \times 10^{-6}$ | $< 3.7 \times 10^{-4}$ [57] | $\lesssim 5.7 \times 10^{-5}$ | $1/3$ |
| $\omega$ | $(1.48 \pm 0.08) \times 10^{-6}$ | [14] | $10^{-5}$ | $8 \times 10^{-7}$ | $< 5.5 \times 10^{-4}$ [55] | $\lesssim 8.2 \times 10^{-5}$ | $1/60$ |
| $\phi$ | $(2.31 \pm 0.11) \times 10^{-6}$ | [14] | $10^{-3}$ | $2 \times 10^{-4}$ | $< 3.0 \times 10^{-4}$ [57] | $\lesssim 4.5 \times 10^{-5}$ | $1/20$ |
| $J/\psi$ | $(3.0 \pm 0.2) \times 10^{-6}$ | [14,59,60] | $0.05\text{–}0.06$ | $8 \times 10^{-4}$ | $< 2.0 \times 10^{-4}$ [58] | $\lesssim 3.9 \times 10^{-5}$ [31] | $1/10$ |
| $\psi(2S)$ | $(1.03 \pm 0.06) \times 10^{-6}$ | [56,60] | $\sim 0.05$ | $\sim 8 \times 10^{-4}$ | $< 9.9 \times 10^{-4}$ [56] | $\lesssim 1.4 \times 10^{-4}$ | $1/140$ |
| $\Upsilon(1S)$ | $(4.6\text{–}30.) \times 10^{-9}$ | [14,59–61] | $0.8\text{–}1$ | $4 \times 10^{-4}$ | $< 2.5 \times 10^{-4}$ [58] | $\lesssim 3.8 \times 10^{-5}$ | $1/10^4$ |
| $\Upsilon(2S)$ | $(1.4\text{–}14.) \times 10^{-9}$ | [14,59–61] | $0.9\text{–}1$ | $3 \times 10^{-4}$ | $< 4.2 \times 10^{-4}$ [58] | $\lesssim 6.4 \times 10^{-5}$ | $1/10^4$ |
| $\Upsilon(3S)$ | $(1.9\text{–}11.) \times 10^{-9}$ | [14,59–61] | $0.9\text{–}1$ | $3 \times 10^{-4}$ | $< 3.4 \times 10^{-4}$ [58] | $\lesssim 5.2 \times 10^{-5}$ | $1/10^4$ |

## 3.2 Higgs decays into a Z boson plus a neutral meson

Higgs decays of the form $H \to Z + M$ involve neutral currents and are very similar to the $H \to \gamma + M$ decays just discussed except that now both pseudoscalar and vector mesons can be produced, and the interference between the direct and indirect amplitudes are smaller, Eqs. (3)–(4). Although these decays are less useful for probing light-quark Yukawa couplings, they are sensitive to the important effective $H\gamma Z$ coupling, and constitute a background for exotic BSM decays of the Higgs boson into, e.g., a Z boson plus an ALP, $H \to Z + a$ [62].

Table 3 compiles the theoretical values and experimental limits for the $H \to Z + M$ branching ratios, and Fig. 6 presents them in graphical form. All decays have rates in the $\mathcal{O}(10^{-5}\text{–}10^{-6})$ range and are dominated by the indirect amplitudes, as indicated by the $\delta_{\mathrm{dir}} \approx 10^{-2}\text{–}10^{-7}$ factors estimated. Experimental bounds have been set for a few channels in the $\mathcal{O}(10^{-2}\text{–}10^{-3})$ range, which are about a factor of ten worse than their $H \to \gamma + M$ counterparts because of

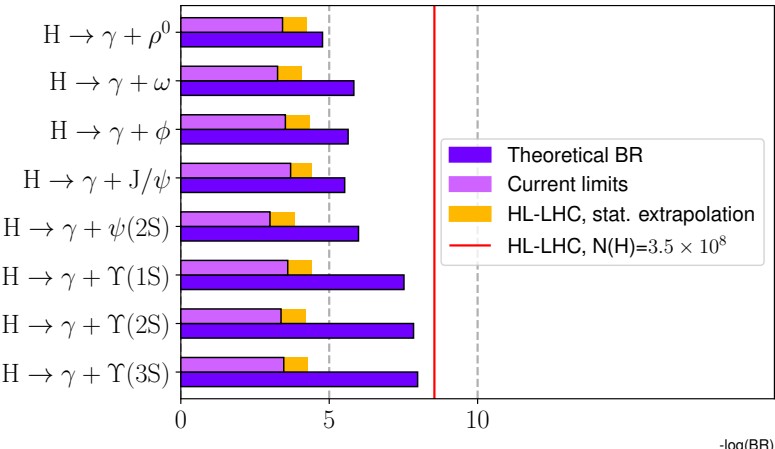

Figure 5: Branching ratios (in negative log scale) of exclusive $H \to \gamma +$ vector-meson decays. Most recent theoretical predictions (blue bars) compared to current experimental limits (violet) and expected conservative HL-LHC bounds (orange). The red vertical line indicates the minimum $\mathcal{B}$ value reachable at the HL-LHC given just by the total number of H bosons expected to be produced.

the extra loss from the $Z \to \ell^+\ell^-$ selection (with $\mathcal{B} \approx 3.4\%$ for each $\ell^+\ell^-$ pair). Conservatively, the HL-LHC will be able to set bounds about 10–70 times above the expected SM values. Bottomonia-plus-Z and $\eta_c$-plus-Z decays have the largest decay rates, $\mathcal{O}(10^{-5})$, but no limit has been set to date for them.

Table 3: Exclusive Higgs decay rates into a Z boson plus a meson. For each decay, we provide the theoretical branching fraction(s), the relative size of the direct contribution to the total branching fraction ($\delta_{\text{dir}}$), the exclusive-direct-to-inclusive decay amplitude ratio, the current experimental limit and that conservatively estimated for HL-LHC (as well as the theory over HL-LHC-expected ratio).

| | Theoretical | | | | Experimental limits | | |
|---|---|---|---|---|---|---|---|
| $H \to Z + M$ | branching fraction | | $\delta_{\text{dir}}$ | $\lvert \mathcal{A}_{\text{dir}}/\mathcal{A}_{H \to q\bar{q}} \rvert$ | current | HL-LHC | $\frac{\mathcal{B}(\text{th})}{\mathcal{B}(\text{exp, HL-LHC})}$ |
| $H \to Z + \pi^0$ | $2.3 \times 10^{-6}$ | [4,54] | $+\mathcal{O}(10^{-7})$ | $\mathcal{O}(10^{-7})$ | – | – | – |
| $\eta$ | $(8.3 \pm 0.9) \times 10^{-7}$ | [54] | | | – | – | – |
| $\rho^0$ | $(7.19 - 14.0) \times 10^{-6}$ | [4,54] | $-\mathcal{O}(10^{-6})$ | $\mathcal{O}(10^{-5})$ | $< 1.2 \times 10^{-2}$ [63] | $\lesssim 1.8 \times 10^{-3}$ | 1/10 |
| $\omega$ | $(5.6 - 16) \times 10^{-7}$ | [4,54] | $-\mathcal{O}(10^{-4})$ | $\mathcal{O}(10^{-6})$ | – | – | – |
| $\eta'$ | $(1.24 \pm 0.13) \times 10^{-6}$ | [54] | | | – | – | – |
| $\phi$ | $(2.4 - 4.2) \times 10^{-6}$ | [4,54] | $-\mathcal{O}(10^{-4})$ | $\mathcal{O}(10^{-5})$ | $< 3.6 \times 10^{-3}$ [63] | $\lesssim 5.4 \times 10^{-4}$ | 1/20 |
| $\eta_c$ | $(1.00 \pm 0.01) \times 10^{-5}$ | [4,64] | $+\mathcal{O}(10^{-4})$ | $\mathcal{O}(10^{-6})$ | – | – | – |
| $J/\psi$ | $(2.3 - 3.4) \times 10^{-6}$ | [54,65,66] | $-\mathcal{O}(10^{-2})$ | $\mathcal{O}(10^{-5})$ | $< 1.9 \times 10^{-3}$ [67] | $\lesssim 2.1 \times 10^{-4}$ [32] | 1/10 |
| $\psi(2S)$ | $1.5 \times 10^{-6}$ | [66] | $-\mathcal{O}(10^{-2})$ | $\mathcal{O}(10^{-5})$ | $< 6.6 \times 10^{-3}$ [67] | $\lesssim 1.0 \times 10^{-3}$ | 1/70 |
| $\eta_b$ | $(2.7 - 4.7) \times 10^{-5}$ | [64,68] | $-\mathcal{O}(10^{-4})$ | $\mathcal{O}(10^{-7})$ | – | – | – |
| $\Upsilon(1S)$ | $(1.54 - 1.7) \times 10^{-5}$ | [54,65,66] | $+(0.01 - 0.04)$ | $\mathcal{O}(10^{-5})$ | – | – | – |
| $\Upsilon(2S)$ | $(7.5 - 8.9) \times 10^{-6}$ | [54,66] | $+\mathcal{O}(10^{-2})$ | $\mathcal{O}(10^{-5})$ | – | – | – |
| $\Upsilon(3S)$ | $(5.63 - 6.7) \times 10^{-6}$ | [54,66] | $+\mathcal{O}(10^{-2})$ | $\mathcal{O}(10^{-5})$ | – | – | – |

## 3.3 Higgs decays into a photon or Z boson plus a neutral flavoured meson

The third diagram of Fig. 4 shows the radiative decay into a neutral gauge boson plus a flavoured meson. In the SM, this FCNC process can only proceed through a W loop (Fig. 7, left), and we are not aware of any theoretical calculation of these rates to date. We have estimated the branching fractions for $H \to \gamma + VM$, $Z + VM$ with $VM = K^{*0}, D^{*0}, B_s^{*0}, B_d^{*0}$, which are all excited flavoured vector mesons, as explained next.

On the one hand, calculations for the inclusive flavour-changing $H \to qq'$ rates (where

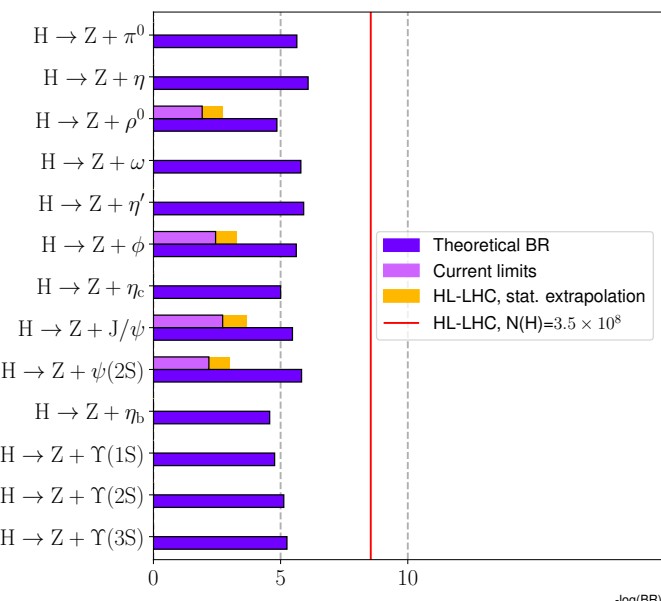

Figure 6: Branching ratios (in negative log scale) of exclusive H → Z + meson decays. The most recent theoretical predictions (blue bars) are compared to current experimental limits (violet) and conservatively expected HL-LHC bounds (orange). The red vertical line indicates the minimum $\mathcal{B}$ value reachable at the HL-LHC given just by the total number of H bosons expected to be produced.

qq′ represents the sum of $q\bar{q}' + q'\bar{q}$ channels) exist [53, 69, 70], but they do not include the gauge boson emission nor the exclusive final-state meson formation. The predicted FCNC Higgs branching fractions, $\mathcal{B}(H \to qq') \equiv \mathcal{B}(H \to q\bar{q}' + q'\bar{q})$, amount to[1]

$$
\begin{aligned}
\mathcal{B}(H \to uc) &= (2.7 \pm 0.5) \cdot 10^{-20}, & \mathcal{B}(H \to db) &= (3.8 \pm 0.6) \cdot 10^{-9}, \\
\mathcal{B}(H \to sb) &= (8.9 \pm 1.5) \cdot 10^{-8}, & \mathcal{B}(H \to ds) &= (1.9 \pm 0.3) \cdot 10^{-15}.
\end{aligned}
\tag{8}
$$

On the other hand, the study [13] determined the H → γ + VM branching fractions, and the work [54] provided the H → Z + VM decay widths assuming arbitrary $\mathcal{O}(1)$ flavour-changing Yukawa couplings. From the results of [13] (resp., [54]), the branching ratios of Higgs decaying into a photon (resp., a Z boson) plus a flavoured neutral meson can be determined through the following EFT + LCDA-based expressions

$$
\mathcal{B}\left(H \to \gamma + VM(qq')\right) = \frac{\alpha(0)}{2\,m_H} \left(\frac{f_{VM} m_{VM}}{2\,\lambda_{VM}(\mu)} Q_q\right)^2 \frac{|\kappa_{qq'}|^2 + |\kappa_{q'q}|^2}{\Gamma_H},
\tag{9}
$$

$$
\mathcal{B}\left(H \to Z + VM(qq')\right) = \frac{9m_H}{8\pi v^2} \left(f_{VM}^{\perp}(\mu_{HZ}) v_q\right)^2 \frac{|\kappa_{qq'}|^2 + |\kappa_{q'q}|^2}{2\,\Gamma_H} \frac{r_Z}{(1-r_Z)^3} (1 - r_Z^2 + 2r_Z \ln r_Z)^2,
\tag{10}
$$

where $\kappa_{qq'}$ are the off-diagonal Higgs coupling to fermions, $Q_q$ the quarks' charge, $\Gamma_H = 4.1$ MeV the total Higgs width, $v = 246$ GeV the Higgs vacuum expectation value, $v_q = T_3^q/2 - Q_q \sin^2 \theta_w$ the vector current coupling of the Z boson, $T_3^q$ is third component of weak isospin, and $r_Z = m_Z^2/m_H^2$ is the squared Z-to-H mass ratio. In Eq. (9), we have used the meson HQET first inverse moments $\lambda_{VM}(\mu)$ [72, 73], with the numerical values $\lambda_{B_s^{*0}} = \lambda_{B_d^{*0}} = \lambda_B$, $\lambda_{D_s^{*0}} = \lambda_D$, and $\lambda_{K^{*0}} = \lambda_K$ quoted in Ref. [16]. In Eq. (10), the $f_{VM}^{\perp}(\mu_{HZ})$ objects are the transverse decay constants of the vector meson, evaluated at the $\mu_{HZ} = (m_H^2 - m_Z^2)/m_H \approx 58.8$ GeV

---

[1]The H → ds branching fraction has been updated as per Ref. [71].

hard scale. These transverse decay constants represent the coupling of the vector meson to the tensor current defined by

$$\langle 0|\overline{q}'(0)\sigma^{\mu\nu}q(0)|VM(p,\lambda)\rangle = if_{\text{VM}}^{\perp}(\mu)(p^{\mu}\varepsilon^{\nu} - p^{\nu}\varepsilon^{\mu}), \tag{11}$$

where $p$, $\lambda$ are the momentum and polarization of the vector meson, respectively, and $\mu$ is the factorization scale. The corresponding numerical values used here are [74,75]: $f_{\text{B}^{*0}}^{\perp}(3 \text{ GeV}) = 0.2$ GeV, $f_{\text{B}_s^{*0}}^{\perp}(3 \text{ GeV}) = 0.236$ GeV, $f_{\text{K}^{*0}}^{\perp}(2 \text{ GeV}) = 0.717 f_{\text{K}^{*0}}$, $f_{\text{D}^{*0}}^{\perp}(2 \text{ GeV}) = 0.202$ GeV. These transverse decay constants can be evolved to the higher scale $\mu_{\text{HZ}}$ using the renormalization group equation [14], giving: $f_{\text{VM}}^{\perp}(\mu_{\text{HZ}}) = 0.86 f_{\text{VM}}^{\perp}(3 \text{ GeV}) = 0.83 f_{\text{VM}}^{\perp}(2 \text{ GeV})$, using $\alpha_{\text{s}}(\mu_{\text{HZ}}) = 0.126$, $\alpha_{\text{s}}(3 \text{ GeV}) = 0.25$, $\alpha_{\text{s}}(2 \text{ GeV}) = 0.29$, and $N_{\text{f}} = 5$ quark flavours.

Finally, the last ingredient needed to compute the $H \to \gamma + \text{VM}^*$, $Z + \text{VM}^*$ branching fractions are the off-diagonal Higgs coupling to fermions $\kappa_{\text{qq}'}$, appearing in the Lagrangian density

$$\mathcal{L}_{\text{H}\to\overline{q}q'} = \sum_q \kappa_{\text{qq}'} h\overline{q}_L q'_R + \text{h.c.} , \tag{12}$$

which can be determined by matching the effective tree-level expressions from the Lagrangian to the NLO branching fractions given by Eq. (8):

$$\mathcal{B}(\text{H} \to qq') \times \Gamma_{\text{H}} = \Gamma(\text{H} \to \overline{q}q' + \overline{q'}q) = \frac{N_c}{8\pi}\left(|\kappa_{\text{qq}'}|^2 + |\kappa_{\text{q}'\text{q}}|^2\right)m_{\text{H}}. \tag{13}$$

The final results for the off-diagonal Higgs couplings read:

$$\begin{cases} |\kappa_{\text{bs}}|^2 + |\kappa_{\text{sb}}|^2 = 2.4 \times 10^{-11}, \\ |\kappa_{\text{bd}}|^2 + |\kappa_{\text{db}}|^2 = 1.0 \times 10^{-12}, \\ |\kappa_{\text{cu}}|^2 + |\kappa_{\text{uc}}|^2 = 7.4 \times 10^{-24}, \\ |\kappa_{\text{ds}}|^2 + |\kappa_{\text{sd}}|^2 = 5.2 \times 10^{-19}. \end{cases} \tag{14}$$

Combining Eq. (14) with (9) or (10), we obtain the theoretical branching fractions listed in Table 4 and plotted in Fig. 7 (right). Such exclusive FCNC radiative decays are extremely suppressed in the SM, in the range of $\mathcal{O}(10^{-14}$–$10^{-30})$. Therefore, radiative flavoured meson decays of the Higgs boson appear utterly rare to be visible at any current or future collider, and therefore a very clean probe of any BSM physics that may enhance Higgs FCNC decays. Experimental limits have been recently set for the $H \to \gamma + \text{K}^{*0}$, $\gamma + \text{D}^{*0}$ channels at $\mathcal{O}(10^{-5})$ [55], to be compared with our $\mathcal{O}(10^{-23})$ SM prediction.

### 3.4 Higgs decays into a W boson plus a charged meson

The charged $H \to \text{W}^{\pm}\text{M}^{\mp}$ decays (diagrams of Fig. 4 with a W boson emitted in the final state) differ qualitatively from the neutral-boson decays discussed above, because the W attaches itself to a charged current, and one can probe flavour-violating couplings of the Higgs boson. Theoretically, the calculations are more complicated as the W has both transverse and longitudinal polarizations, yielding lengthier analytical expressions [12,54]. Table 5 lists the theoretical predictions for Higgs decays into a $\text{W}^{\pm}$ boson plus a charged meson, and Fig. 8 presents them in graphical form. The rates for these processes range roughly between $10^{-5}$ and $10^{-10}$ and are dominated by the indirect amplitudes, as indicated by the $\delta_{\text{dir}} \approx 10^{-3}$–$10^{-6}$ factors quoted. The EFT + NRQM theoretical predictions of Ref. [4], which update the $\mathcal{B}$ numerical values computed in [12] (but have likely a typo in the $\mathcal{B}(\text{H} \to \text{W}^{\mp} + \text{B}^{*\pm}) = 10^{-5}$ rate quoted, which should be $10^{-10}$), agree with the alternative EFT + LCDA rates of Ref. [54]. There has been no experimental search to date at the LHC for any of these 11 decays. Experimentally, the most promising channel for HL-LHC, with a $\mathcal{O}(10^{-5})$ branching fraction, is $H \to \text{W}^{\pm} + \rho^{\mp}$ with the rho meson decaying into two pions with almost 100% probability.

Table 4: Exclusive Higgs decay rates into a neutral gauge boson plus a flavoured meson. For each decay, we provide the branching fraction derived via Eqs. (9)–(10), as well as the current experimental limit and that conservatively estimated for HL-LHC (as well as the SM over HL-LHC-expected ratio).

| $H \to X + M$ | Theoretical branching fraction | | Experimental limits current | HL-LHC | $\frac{\mathcal{B}(\text{th})}{\mathcal{B}(\text{exp, HL-LHC})}$ |
|---|---|---|---|---|---|
| $H \to \gamma + K^{*0}$ | $2.6 \times 10^{-23}$ | (this work) | $< 2.2 \times 10^{-4}$ [55] | $\lesssim 3.3 \times 10^{-5}$ | $1/10^{18}$ |
| $D^{*0}$ | $6.7 \times 10^{-27}$ | (this work) | $< 1.0 \times 10^{-3}$ [76] | $\lesssim 1.5 \times 10^{-4}$ | $1/10^{22}$ |
| $B^{*0}$ | $8.2 \times 10^{-16}$ | (this work) | – | – | – |
| $B_s^{*0}$ | $1.8 \times 10^{-14}$ | (this work) | – | – | – |
| $H \to Z + K^{*0}$ | $2.2 \times 10^{-25}$ | (this work) | – | – | – |
| $D^{*0}$ | $1.8 \times 10^{-30}$ | (this work) | – | – | – |
| $B^{*0}$ | $2.4 \times 10^{-19}$ | (this work) | – | – | – |
| $B_s^{*0}$ | $2.9 \times 10^{-17}$ | (this work) | – | – | – |

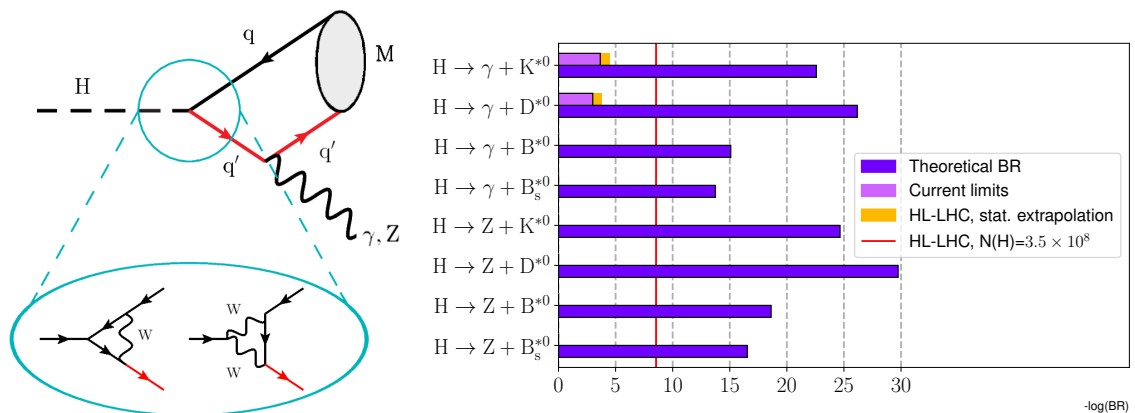

Figure 7: Left: Schematic diagram of exclusive flavour-changing Higgs decays into a meson plus a photon or a Z boson. The solid fermion lines represent quarks, the wavy ones bosons, and the gray blob a neutral meson. Right: Branching fractions (in negative log scale) of exclusive $H \to \gamma + VM^*$, $Z + VM^*$ decays, where $VM^*$ are excited flavoured neutral mesons. The SM predictions (blue bars) from Eqs. (9) and (10) are compared with current experimental limits (violet) and expected conservative HL-LHC bounds (orange). The red vertical line indicates the minimum $\mathcal{B}$ value reachable at the HL-LHC given just by the total number of H bosons expected to be produced.

Table 5: Exclusive Higgs decay rates into a W boson plus a meson. For each decay, we provide the theoretical branching fraction(s), and the relative size of the direct contribution to the total branching fraction ($\delta_{\text{dir}}$). No current experimental limits, nor future estimates for them, exist.

| H → W | + | M | Theoretical branching fraction | | $\delta_{\text{dir}}$ | Experimental limits | |
|---|---|---|---|---|---|---|---|
| | | | | | | current | HL-LHC |
| H → W$^{\mp}$ | + | $\pi^{\pm}$ | $(4.3 \pm 0.3)$ | $\times 10^{-6}$ [4,54] | $+\mathcal{O}(10^{-6})$ | – | – |
| | | $\rho^{\pm}$ | $(1.3 \pm 0.3)$ | $\times 10^{-5}$ [4,54] | $+\mathcal{O}(10^{-6})$ | – | – |
| | | K$^{\pm}$ | $(3.3 \pm 0.1)$ | $\times 10^{-7}$ [4,54] | $+\mathcal{O}(10^{-5})$ | – | – |
| | | K$^{*\pm}$ | $(5.0 \pm 1.0)$ | $\times 10^{-7}$ [4,54] | $+\mathcal{O}(10^{-5})$ | – | – |
| | | D$^{\pm}$ | $(5.7 \pm 0.7)$ | $\times 10^{-7}$ [4,54] | $+\mathcal{O}(10^{-5})$ | – | – |
| | | D$^{*\pm}$ | $(1.2 \pm 0.3)$ | $\times 10^{-6}$ [4,54] | $+\mathcal{O}(10^{-3})$ | – | – |
| | | D$_s^{\pm}$ | $(1.6 \pm 0.2)$ | $\times 10^{-5}$ [4,54] | $+\mathcal{O}(10^{-5})$ | – | – |
| | | D$_s^{*\pm}$ | $(3.0 \pm 0.7)$ | $\times 10^{-5}$ [4,54] | $+\mathcal{O}(10^{-3})$ | – | – |
| | | B$^{\pm}$ | $(1.6 \pm 0.5)$ | $\times 10^{-10}$ [4,54] | $+\mathcal{O}(10^{-4})$ | – | – |
| | | B$^{*\pm}$ | $(1.4 \pm 0.4)$ | $\times 10^{-10}$ [4,54] | $+\mathcal{O}(10^{-3})$ | – | – |
| | | B$_c^{\pm}$ | $(5.0 \pm 4.0)$ | $\times 10^{-8}$ [4,54] | $+\mathcal{O}(10^{-3})$ | – | – |

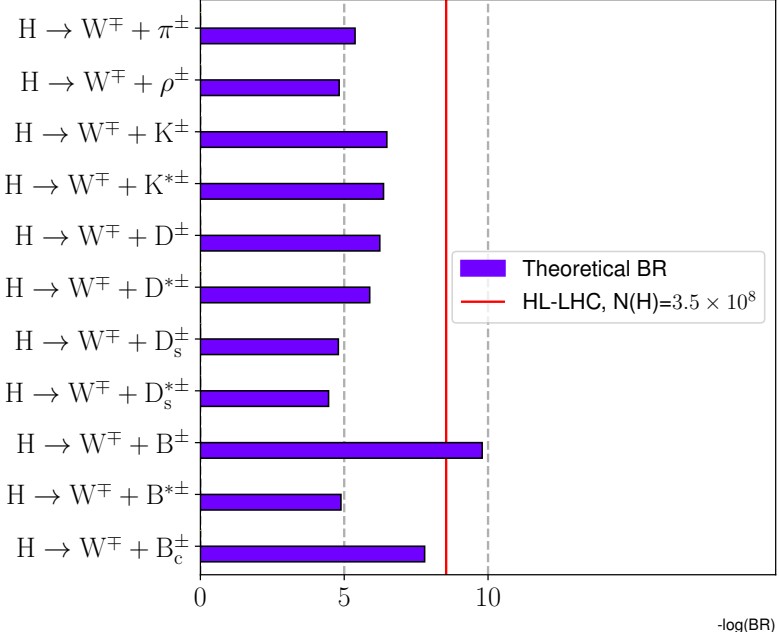

Figure 8: Theoretical branching fractions (blue bars, in negative log scale) of exclusive $H \to W^{\pm} +$ meson decays. No experimental limits exist to date. The red vertical line indicates the minimum $\mathcal{B}$ value reachable at the HL-LHC given just by the total number of H bosons expected to be produced.

# 4 Radiative Higgs leptonium decays

Figure 9 shows the diagrams of the Higgs decay into a photon or a Z boson plus a lepton-antilepton bound state $(\ell^+\ell^-)$, where $(\ell^+\ell^-) = (e^+e^-), (\mu^+\mu^-), (\tau^+\tau^-)$ represent positronium, true muonium [77], and true tauonium [78], respectively. Depending on the accompanying gauge boson (Z or $\gamma$), leptonium can be produced in spin triplet (ortho) and/or spin singlet (para) states. Such decays are similar to the exclusive radiative decays into mesons shown in Fig. 4, but changing the quarks' for leptons' lines. Of course, in the SM there is no unlike-flavour leptonic decay possible, at variance with the $H \to W^\pm + M^\mp$ case. However, in the presence of BSM physics leading to LFV Higgs decays, $(\ell^\pm\ell'^\mp)$ bound states could be formed directly in exclusive radiative decays. We provide estimates of the SM decay rates obtained here by considering the similar $H \to \gamma + M$, $Z + M$ processes and changing the quarkonium form-factors by leptonium ones, as well as from the more detailed calculations of Ref. [79]. If such decays were to be produced with relatively large rates, they would provide interesting cross checks for any LFUV effects potentially observed with the "open" leptons [80].

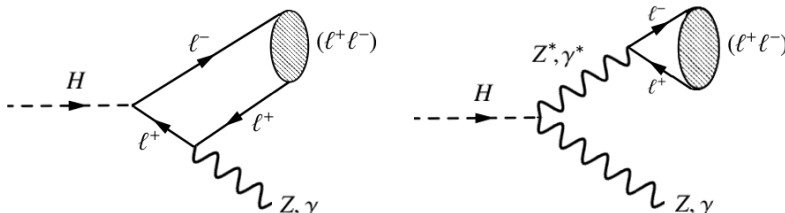

Figure 9: Schematic diagrams of exclusive H boson decays into a photon or a Z boson plus a leptonium state in the direct (left) and indirect (right) channels. The solid fermion lines represent leptons, the gray blob represents the $(\ell^+\ell^-)$ bound state.

The overall probability for forming an onium bound state is determined by one single parameter: its radial wavefunction at the origin, which for leptonium bound states (of constituent lepton mass $m_\ell$ and principal quantum number $n$) reads [78],

$$|\phi_{n,(\ell\ell)}(r=0)|^2 = \frac{(m_\ell \alpha(0))^3}{8\pi n^3}. \tag{15}$$

Since the QED coupling is much smaller than the QCD one, $\alpha(m_{\ell\ell}) \ll C_F \alpha_s(m_{q\bar{q}})$ with colour factor $C_F = 4/3$, and since the charged lepton masses are smaller than the quark masses of the same generation, $m_{\ell\ell} \ll m_{q\bar{q}}$, the ratio $[\alpha(0)m_{\ell\ell}/(\alpha_s(m_{q\bar{q}})m_{q\bar{q}})]^3$ is very small, and one can anticipate that those decays will be orders-of-magnitude more suppressed than the $H \to \gamma + M(q\bar{q})$ ones discussed in Section 3.

We calculate first the $H \to \gamma + (\ell^+\ell^-)$ decay width where, due to charge-conjugation conservation, the leptonium can only be in the ortho-state, $(\ell^+\ell^-)_1$. The branching fraction of this decay can be derived from the similar expression for radiative quarkonium meson decays, and reads

$$\mathcal{B}(H \to \gamma + (\ell^+\ell^-)_1) = \frac{1}{8\pi} \frac{m_H^2 - m_{\ell\ell}^2}{m_H^2 \Gamma_H} |\mathcal{A}_{dir} + \mathcal{A}_{ind}|^2, \tag{16}$$

with the direct and indirect amplitudes corresponding, respectively, to the left and right diagrams of Fig. 9, and where the $H \to \gamma + Z^*$ contribution followed by the $Z^* \to (\ell^+\ell^-)_1$ transition is negligible compared to the corresponding $H \to \gamma + \gamma^*$ one with partial width

$\Gamma_{H\to\gamma\gamma} = 2.5 \cdot 10^{-3}\,\Gamma_H$. These amplitudes can be written as a function of the leptonium wavefunction at the origin, $\phi_{n,(\ell\ell)}(0)$, as follows

$$\mathcal{A}_{\text{dir}} = 2Q_\ell \sqrt{4\pi\alpha(0)} \left(\sqrt{2}G_F\, m_{\ell\ell}\right)^{1/2} \frac{m_H^2 - m_{\ell\ell}^2}{\sqrt{m_H}(m_H^2 - m_{\ell\ell}^2/2 - 2m_\ell^2)}\,\phi_{n,(\ell\ell)}(0), \tag{17}$$

$$\mathcal{A}_{\text{ind}} = \frac{Q_\ell \sqrt{4\pi\alpha(0)}\, f_{\ell\ell}}{m_{\ell\ell}} \left(16\pi\Gamma_{H\to\gamma\gamma}\right)^{1/2} \frac{m_H^2 - m_{\ell\ell}^2}{m_H^2} \tag{18}$$

$$= -\frac{2Q_\ell \sqrt{4\pi\alpha(0)}}{m_{\ell\ell}^{3/2}} \left(16\pi\Gamma_{H\to\gamma\gamma}\right)^{1/2} \frac{m_H^2 - m_{\ell\ell}^2}{m_H^2}\,\phi_{n,(\ell\ell)}(0), \tag{19}$$

where for the last $\mathcal{A}_{\text{ind}}$ equality, we have adopted the equivalent of the Van Royen–Weisskopf formula for mesons [81,82] to relate the leptonium decay constant, $f_{\ell\ell}$, to its wavefunction at the origin: $f_{\ell\ell}^2 = 4\frac{|\phi_{M,\ell\ell}(0)|^2}{m_{\ell\ell}}$.

We choose $\phi_{\ell\ell}(0)$ to be real and the phase for the decay constant $f_{\ell\ell}$, which decides the (destructive) interference between indirect and direct decay amplitudes, to be positive, as in the quarkonium case [11]. Since $m_{\ell\ell} \ll m_H$, plugging Eq. (15) into (19), one can see that the indirect amplitudes are independent of the leptonium masses. Using Eqs. (16)–(19), we determine the radiative ortholeptonium ($n = 1$) branching fractions listed in the first three rows of Table 6. The radiative leptonium decays of the Higgs boson have all numerically similar $\mathcal{O}(10^{-12})$ rates for the three leptons. The experimental search has a very clean signature characterized by a secondary vertex from the boosted $(\ell^+\ell^-)_1$ decay, which leads to significantly displaced triphotons (from positronium and dimuonium decays), $e^+e^-$ pairs (from positronium breakup[2] and from direct dimuonium and ditauonium decays), or $\mu^+\mu^-$ pairs (from direct ditauonium decays). No search has been performed to date at the LHC, and only a future machine like FCC-hh can try to set limits on them at about 10 times their SM values.

We consider next the $H \to Z + (\ell^+\ell^-)_{0,1}$ decay rates where, at variance with the photon case, scalar (para-) and vector (ortho-) leptonium states can be produced. The $Z + (\ell^+\ell^-)_1$ decay width can be written as

$$\Gamma(H \to Z + (\ell^+\ell^-)_1) = \Gamma_1 + \Gamma_2 + \Gamma_3, \tag{20}$$

where $\Gamma_1$, $\Gamma_2$, and $\Gamma_3$ are the contributions from $H \to Z + Z^* \to Z + (\ell^+\ell^-)_1$, $H \to Z + \gamma^* \to Z + (\ell^+\ell^-)_1$ decays, and their interference, respectively. Details on the numerical evaluation of the three terms of Eq. (20) can be found in Ref. [16]. The rates for $H \to Z + (\ell^+\ell^-)_1$ decays amount to $\mathcal{O}(10^{-11}$–$10^{-13})$ (second three rows of Table 6), and have not been searched-for to date at the LHC. A recent work [79] has computed higher-order corrections to these decays, finding consistent results with ours except for the $H \to \gamma + (e^+e^-)_1$ channel that would have about 25 times larger decay rates. In this case, a high-luminosity machine such as FCC-hh would be able to provide limits approaching the SM value.

Lastly, the rates of the $H \to Z + (\ell^+\ell^-)_0$ decays can be obtained from

$$\Gamma(H \to Z + (\ell^+\ell^-)_0) = \frac{m_H^3}{4\pi v^4}\,\lambda^{3/2}(1, r_Z, r_{\ell\ell}) \left|F^{Z+(\ell\ell)_0}\right|^2 \text{ with } F^{Z+(\ell\ell)_0} = F_{\text{dir}}^{Z+(\ell\ell)_0} + F_{\text{ind}}^{Z+(\ell\ell)_0}, \tag{21}$$

with the direct amplitude contribution suppressed by a factor $m_{\ell\ell}^2/m_H^2$ or $m_\ell^2/m_H^2$, that makes it completely negligible (amounting to a maximum of 0.1% in the $(\tau\tau)_0$ case). The contribution

---

[2]The boosted positronium would have an extremely long path-length (its natural triphoton decay, with lifetime $c\tau = 142$ ns, would take place at $L = (c\tau)(\beta * \gamma) \approx 1450$ km, for $\beta\gamma \approx (m_H - m_Z)/m_{(ee)} \approx 3.4 \cdot 10^4$) and would appear instead as missing transverse energy, but the bound state could be first potentially broken into its individual $e^+e^-$ components by interactions with the detector material and/or magnetic field [83].

from the indirect amplitude to the width (21) is given by $F_{\text{ind}}^{Z+(\ell\ell)_0} = f_{\ell\ell}\, a_\ell$ with $a_\ell = T_3^\ell/2 = 1/4$, and results in Z-plus-paraleptonium decay rates, $H \rightarrow Z + (\ell^+\ell^-)_0$, which amount to $10^{-12}$–$10^{-16}$ (listed in the three last rows of Table 6).

All branching fractions computed here for Higgs decays into Z or $\gamma$ plus leptonium are plotted in Fig. 10. As discussed above, the decay rates are minuscule, in the $\mathcal{O}(10^{-10}$–$10^{-16})$ range, and only BSM effects enhancing them would make them visible. A machine producing as many Higgs bosons as the FCC-hh can attempt to observe the $H \rightarrow \gamma + (3\gamma)$ with a displaced triphoton vertex from the late orthopositronium $(ee)_1 \rightarrow 3\gamma$ decay, or $H \rightarrow \gamma + e^+e^-$ with a displaced dielectron from the detector- or magnetic-field-induced breaking of the bound state into its constituents.

Table 6: Exclusive Higgs decay rates to a photon or a Z boson plus an ortho- $(\ell^+\ell^-)_1$ or para- $(\ell^+\ell^-)_0$ leptonium state (only the ground states, $n = 1$, are considered). For each decay, we provide the theoretical SM branching fraction predictions. No current experimental limits, nor future estimates for them, exist.

| $H \rightarrow V$ | $+$ | $(\ell\ell)$ | Theoretical branching fraction | | Experimental limits current | HL-LHC |
|---|---|---|---|---|---|---|
| $H \rightarrow \gamma$ | $+$ | $(ee)_1$ | 3.5–88 $\times 10^{-12}$ | (this work), [79] | – | – |
| | | $(\mu\mu)_1$ | 3.5–11.2 $\times 10^{-12}$ | (this work), [79] | – | – |
| | | $(\tau\tau)_1$ | 2.2–3.5 $\times 10^{-12}$ | (this work), [79] | – | – |
| $H \rightarrow Z$ | $+$ | $(ee)_1$ | 5.2–7.9 $\times 10^{-13}$ | (this work), [79] | – | – |
| | | $(\mu\mu)_1$ | 5.7–9.8 $\times 10^{-13}$ | (this work), [79] | – | – |
| | | $(\tau\tau)_1$ | 1.4–5.7 $\times 10^{-11}$ | (this work), [79] | – | – |
| | | $(ee)_0$ | 2.7 $\times 10^{-16}$ | (this work) | – | – |
| | | $(\mu\mu)_0$ | 1.1 $\times 10^{-14}$ | (this work) | – | – |
| | | $(\tau\tau)_0$ | 3.2 $\times 10^{-12}$ | (this work) | – | – |

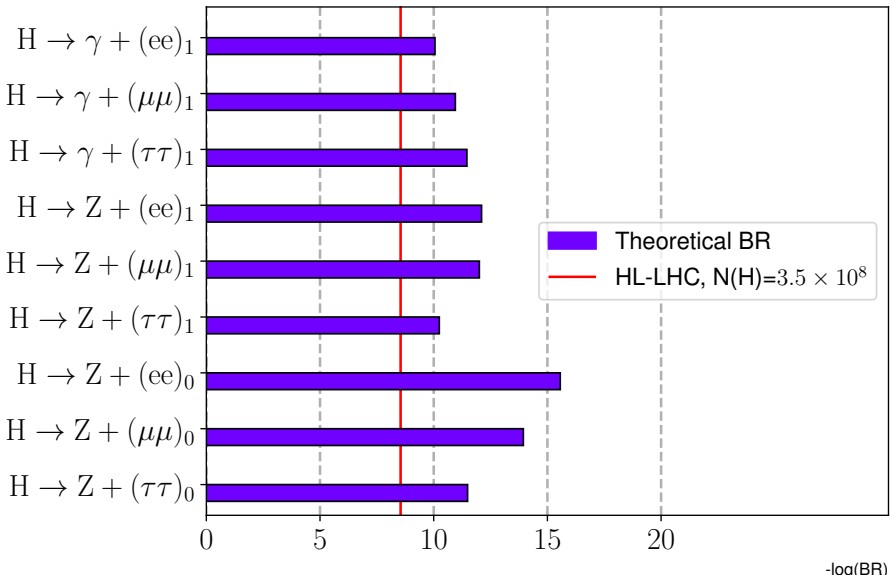

Figure 10: Theoretical branching fractions (blue bars, in negative log scale) of exclusive $H \rightarrow \gamma, Z +$ leptonium decays. No experimental limits exist to date. The red vertical line indicates the minimum $\mathcal{B}$ value reachable at the HL-LHC given just by the total number of H bosons expected to be produced.

# 5 Exclusive Higgs decays into a pair of mesons

Figure 11 shows representative diagrams of the exclusive decay of the Higgs boson into two mesons, that can proceed through a multitude of intermediate states coupling to the scalar particle: quarks, gluons, virtual EW gauge bosons. First estimates of these processes were performed ignoring the internal motion of the $q\bar{q}$ pairs [84], and then further improved within different approaches for the meson-pair formation [85–90]. Except for a channel involving the $\phi$ meson, only exclusive decays into charmonium and/or bottomonium final states have been computed to date, i.e., no calculations of decays to a pair of light mesons exist to our knowledge. The work [79] also computed the double-leptonium decay rates, $H \rightarrow (\ell\ell)_1 + (\ell\ell)_1$, which are further suppressed compared to the radiative leptonium rates discussed in Section 4, have branching fractions in the $\mathcal{O}(10^{-20})$ range, and not considered hereafter.

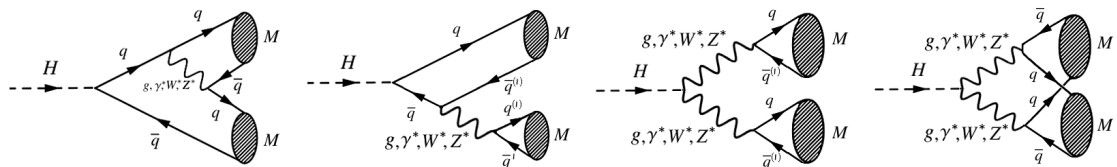

Figure 11: Schematic diagrams of exclusive decays of the H boson into two mesons. The wavy lines indicate gauge bosons, the solid fermion lines represent quarks and the gray blobs are the meson bound states.

Table 7 lists the corresponding theoretical predictions and experimental limits for concrete $H \rightarrow 2(Q\bar{Q})$ and $H \rightarrow (Q\bar{Q})(Q\bar{Q}')$ decay modes. The calculations are carried out in multiple frameworks and predict rates in the $\mathcal{O}(10^{-9}–10^{-11})$ range, with some differences in the results for the same decay because only a subset of the diagrams shown in Fig. 11 has been considered. As a matter of fact, no existing prediction has consistently included all the processes shown in the figure (e.g., the W-induced and quark "crossed" decays are often considered subleading and not added to the rates). The $H \rightarrow \phi + J/\psi$ decay is the only process computed to date that includes the $H \rightarrow W^*W^*$ intermediate diagram. The direct (quark-induced) contributions are only relevant for the heavier double bottomonia with large Yukawa couplings, whereas calculations of final decays involving charmonium ignore them.

Table 7: Exclusive Higgs decay rates to a pair of mesons. For each decay, we list the theoretical branching fraction(s), the current experimental limit and the conservative bounds estimated for HL-LHC (as well as the theory over HL-LHC-expected ratio).

| | | | Theoretical | | Experimental limits | | $\frac{\mathcal{B}(\text{th})}{\mathcal{B}(\text{exp, HL-LHC})}$ |
|---|---|---|---|---|---|---|---|
| H $\rightarrow$ | M | + M | branching fraction | | current | HL-LHC | |
| H $\rightarrow$ | $\phi$ | + $J/\psi$ | $1.0 \times 10^{-9}$ | [85] | – | – | – |
| | $J/\psi$ | + $J/\psi$ | $(1.5–60) \times 10^{-10}$ | [85,87–90] | $< 3.8\times 10^{-4}$ [67] | $\lesssim 5.8 \times 10^{-5}$ | $1/10^4$ |
| | $\psi(2S)$ | + $J/\psi$ | $\sim 5 \times 10^{-11}$ | – | $< 2.1\times 10^{-3}$ [67] | $\lesssim 3.2 \times 10^{-4}$ | $1/10^7$ |
| | $\psi(2S)$ | + $\psi(2S)$ | $(5.1 \pm 2.0) \times 10^{-11}$ | [88] | $< 3.0\times 10^{-3}$ [67] | $\lesssim 4.5 \times 10^{-4}$ | $1/10^7$ |
| | $B_c^{\mp}$ | + $B_c^{\pm}$ | $(2.0–3.0) \times 10^{-10}$ | [86] | – | – | – |
| | $B_c^{*\mp}$ | + $B_c^{*\pm}$ | $(1.4–1.7) \times 10^{-10}$ | [86] | – | – | – |
| | $\Upsilon(1S)$ | + $J/\psi$ | $(0.16–3.6) \times 10^{-10}$ | [85,90] | – | – | – |
| | $\Upsilon(1S)$ | + $\Upsilon(1S)$ | $(1.8–23) \times 10^{-10}$ | [85,87–90] | $< 1.7\times 10^{-3}$ [67] | $\lesssim 2.6 \times 10^{-4}$ | $1/10^5$ |
| | $\Upsilon(2S)$ | + $\Upsilon(2S)$ | $(1.0 \pm 0.2) \times 10^{-10}$ | [88] | – | – | – |
| | $\Upsilon(3S)$ | + $\Upsilon(3S)$ | $(5.7 \pm 1.2) \times 10^{-11}$ | [88] | – | – | – |
| | $\Upsilon(mS)$ | + $\Upsilon(nS)$ | – | – | $< 3.5\times 10^{-4}$ [67] | $\lesssim 9.2 \times 10^{-6}$ [32] | – |

Experimentally, the double-$(Q\bar{Q})$ and $(Q\bar{Q})(Q\bar{Q}')$ decays have been searched for at the LHC [67], but the current $\mathcal{O}(10^{-3}–10^{-4})$ limits are 4–5 orders-of-magnitude larger than the

SM predictions. Their observation at HL-LHC appears unfeasible (Fig. 12), and only a machine like FCC-hh will have the Higgs production rates required to reach those decay modes.

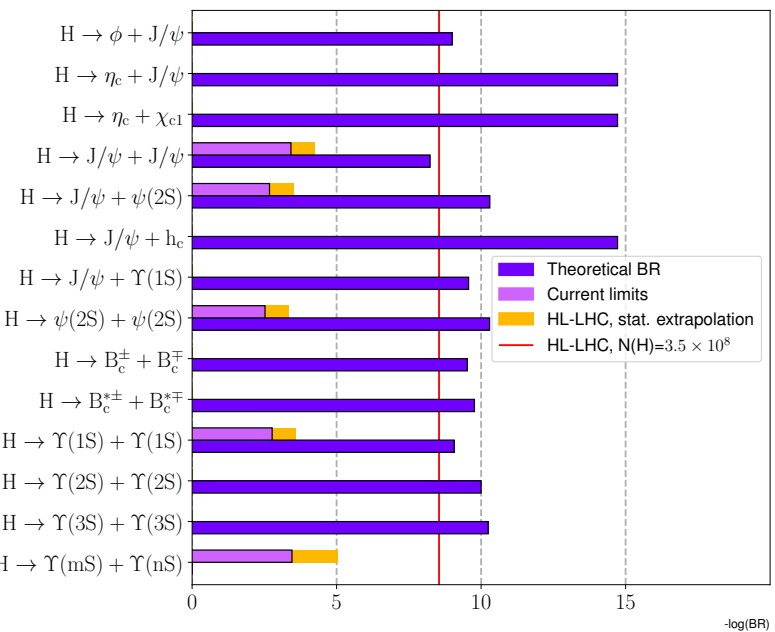

Figure 12: Branching ratios (in negative log scale) of exclusive H → meson + meson decays. Most recent theoretical predictions (blue bars) are compared to current experimental limits (violet) and expected conservative HL-LHC bounds (orange). The red vertical line indicates the minimum $\mathcal{B}$ value reachable at the HL-LHC given just by the total number of H bosons expected to be produced.

# 6 Summary

We have presented a comprehensive survey of the theoretical and experimental status of about 70 rare and exclusive few-body decays of the Standard Model Higgs. Rare decays are defined here as those having branching fractions below $\mathcal{B} \approx 10^{-5}$, and we focus on those with two-to four-particles in the final state. Such decay processes remain experimentally unobserved and only limits have been set for about 20 of them. The study of these decay processes provides a useful window into physics beyond the Standard Model (BSM) either directly, by probing SM suppressed or forbidden processes (such as flavour-changing neutral currents FCNC, lepton-flavour or lepton-flavour-universality violating processes, or spin-selection-rules violating decays), or indirectly as SM backgrounds to multiple exotic BSM Higgs decays (e.g., into axion-like particles ALPs, or dark photons). Additionally, such decays offer a unique opportunity to probe the light-quark Yukawa couplings, and help improve our understanding of quantum chromodynamics (QCD) factorization with small nonperturbative corrections.

We have systematically collected and organized in tabular form the theoretical branching fractions of about 70 rare decay channels, while providing their current experimental limits. Among those, we have estimated for the first time the rates of about 20 new processes including ultrarare Higgs boson decays into photons and/or neutrinos (with $10^{-12}$–$10^{-40}$ rates) and into Z bosons plus gluons or photons (with $10^{-6}$, $10^{-9}$ rates), radiative H boson decays into leptonium states (with rates $10^{-10}$–$10^{-23}$), and exclusive radiative H boson quark-flavour-changing decays (with $10^{-14}$–$10^{-27}$ rates).

Secondly, the feasibility of measuring each of these unobserved decays has been estimated for p-p collisions at the High-Luminosity Large Hadron Collider (HL-LHC). From the number of H bosons expected to be produced at the HL-LHC, $N_{\text{HL-LHC}}(\text{H}) \approx 3.5 \cdot 10^8$, and by statistically extrapolating the current 95% confidence-level limits set for many channels, we provide conservative estimates of the experimental bounds (or observations) achievable at HL-LHC. Among those, in Table 8 we have selected seven interesting decays that can be observed and/or deserve further experimental study in p-p collisions at the LHC. The last column indicates the ratio of theoretical to the experimental rates conservatively extrapolated for HL-LHC, $\mathcal{B}(\text{th})/\mathcal{B}(\text{exp})$.

Table 8: Selection of rare and exclusive SM Higgs decays potentially observable in pp(14 TeV) collisions at the HL-LHC (or for which a limit could be "easily" set from existing data). For each decay, we list the theoretical branching fraction(s), the current experimental limit and the conservative bounds estimated for HL-LHC (as well as the theory over HL-LHC-expected ratio, $\mathcal{B}(\text{th})/\mathcal{B}(\text{exp,HL-LHC})$).

| Decay | | | Theoretical branching fraction | Experimental limits current | HL-LHC | $\frac{\mathcal{B}(\text{th})}{\mathcal{B}(\text{exp, HL-LHC})}$ |
|---|---|---|---|---|---|---|
| | | $\gamma\gamma\gamma\gamma$ | $5.4 \times 10^{-12}$ | – | – | – |
| | $\gamma$ + | $\rho^0$ | $(1.68 \pm 0.08) \times 10^{-5}$ | $< 3.7 \times 10^{-4}$ [57] | $\lesssim 5.7 \times 10^{-5}$ | $\sim 1/4$ |
| | | $\text{J}/\psi$ | $(3.0 \pm 0.2) \times 10^{-6}$ | $< 2.0 \times 10^{-4}$ [58] | $\lesssim 3.9 \times 10^{-5}$ [31] | $\sim 1/10$ |
| H $\to$ | $\text{Z}$ + | $\rho^0$ | $(7.19-14.0) \times 10^{-6}$ | $< 1.2 \times 10^{-2}$ [63] | $\lesssim 1.8 \times 10^{-3}$ | $\sim 1/100$ |
| | | $\Upsilon(1\text{S})$ | $(1.54-1.7) \times 10^{-5}$ | – | – | – |
| | $\text{W}^{\mp}$ + | $\rho^{\pm}$ | $(1.5 \pm 0.1) \times 10^{-5}$ | – | – | – |
| | | $\text{D}_s^{*\pm}$ | $(3.5 \pm 0.2) \times 10^{-5}$ | – | – | – |

The first rare decay listed, H $\to$ 4$\gamma$, has not been directly searched for at the LHC so far, but limits exist on the process H $\to$ a($\gamma\gamma$)a($\gamma\gamma$) with two intermediate ALPs decaying into photons [49–52]. Although its $\mathcal{O}(5 \cdot 10^{-12})$ rate makes its observation impossible at the LHC in the absence of enhancing BSM effects, it would be interesting for ATLAS/CMS to recast any current and future similar ALP searches into lower bounds on the H $\to$ 4$\gamma$ "continuum" decay. The six other decays listed have relatively large rates, $\mathcal{O}(10^{-5})$, with their measurement having been attempted or not to date. The exclusive H $\to \gamma + \rho$ and H $\to \gamma + \text{J}/\psi$ decays, with conservative HL-LHC rates extrapolated at about four and ten times their expected SM values, $\mathcal{B}(\text{th}) \approx 2 \cdot 10^{-5}, 3 \cdot 10^{-6}$, respectively, are listed first. With realistic improvements in the analyses, evidence or observation of these exclusive channels appears at reach. Other interesting exclusive decays indicated are H $\to \text{Z} + \rho$ and H $\to \gamma + \Upsilon$. The first one, has current limits conservatively extrapolated to 100 times its SM rate, whereas the second one has not been searched to date but should be in the same ballpark. Last but not least, we tabulate the most probable exclusive decays of a Higgs boson to a $\text{W}^{\pm}$ boson plus a vector meson ($\rho^{\mp}$, $\text{D}_s^{*\mp}$), whose measurement could be attempted at the HL-LHC.

Of course, the selection of Table 8 is driven by the SM rates and their potential visibility at the HL-LHC, but as explained in this work there are many other suppressed decays of the four heaviest particles that can be enhanced in multiple BSM scenarios, and should be an active part of target searches in the next years. We hope that this document can help guide and prioritize upcoming experimental and theoretical studies of rare and exclusive few-body decays of the Higgs boson, as well as further motivate BSM searches, at the LHC and future colliders.

## Acknowledgements

Support from the EU STRONG-2020 project under the program H2020-INFRAIA-2018-1 grant agreement No. 824093 is acknowledged.

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
