# Peer review of "Rare few-body decays of the Standard Model Higgs boson"

_SciPost Physics Community Reports_

## Round 1 · Referee Report · Anonymous (Referee 1) · 2025-10-24

Strengths

  • comprehensive overview of rare Higgs decays
  • predictions for so far unknown decay rates

Report

The article presents an overview of rare Higgs decays in the SM. Existing results in the literature are reviewed and complemented by new calculations of so far unknown decay rates.

The article is comprehensive, useful for the wider community, well-structured and well-written. As intended by the authors, it will serve as a guide for future experimental and theoretical studies. Therefore, I do not hesitate to recommend the publication of the article as a SciPost community report.

Requested changes

In the text above Fig. 3, the authors could explain briefly why they use the SMEFT@NLO UFO model for a SM calculation. I guess that they rely on an effective coupling not available in the SM model file, but this is not clear from the text.

Recommendation

Publish (easily meets expectations and criteria for this Journal; among top 50%)

  • validity: high
  • significance: good
  • originality: good
  • clarity: good
  • formatting: good
  • grammar: excellent

Author:  Van Dung Le  on 2025-11-27  [id 6085]

(in reply to Report 1 on 2025-10-24)

REFEREE: In the text above Fig. 3, the authors could explain briefly why they use the SMEFT@NLO UFO model for a SM calculation. I guess that they rely on an effective coupling not available in the SM model file, but this is not clear from the text.

ANSWER: Yes, the computation of these two widths relies on the loop-induced $\rm HZ\gamma$ and $Zgg$ processes, for which the SMEFTatNLO framework provides a pointlike effective vertex that facilitate the calculation of those widths. To clarify this point better, the text has been updated as follows:

OLD:

with MG5@NLO using the SMEFT@NLO model~\cite{Degrande:2020evl}.

NEW:

with MG5@NLO using the encoded loop-induced SM effective couplings of the SMEFT@NLO model~\cite{Degrande:2020evl}.

---

## Round 1 · Referee Report · Anonymous (Referee 2) · 2025-10-30

Strengths

1. Complete review of SM predictions for rare Higgs decays
2. Useful reference for future theory improvements and experimental searches

Weaknesses

1. Only very few of the decay modes, collected in Table 8, have SM rates relevant for HL-LHC.

Report

This is a review article on SM predictions for rare Higgs decays, defined as Higgs decays with branching ratios smaller than 10^-5. Results already available in the literature are collected and discussed and, where missing, a prediction is derived using either numerical tools or explicit calculations.

Computations of the decay rates into neutrinos and neutral gauge bosons are performed with MadGraph5_aMC@NLO. Exclusive decays into gauge bosons plus mesons employ QCD factorisation to compute the relevant hadronic matrix elements. Most results are reviewed from the literature, except for decays into a photon or Z boson plus a flavoured neutral meson, that are computed for the first time.

The manuscript is well written and very clear. The results will help in guiding future experimental searches.
The article meets this journal’s acceptance criteria, but there are some minor points that should be addressed before publication.

Requested changes

1. At the end of page 2 it it stated that precision tests of suppressed/forbidden processes in the SM have been mostly studied in B-decays so far. This is evidently not true, as even more powerful constraints on the scale of new physics are obtained from rare/forbidden Kaon decays, LFV tests in muon or tau decays, electric dipole moments, etcetera, etcetera. The sentence should be modified.
2. I don’t understand why the SM prediction for H > nu \bar{nu} is not exactly zero. In the SM, neutrinos are massless, and the decay rate of a scalar into two massless fermions of opposite helicity (as nu and \bar{nu} have) vanish due to angular momentum conservation (the rate is proportional to the fermion mass, as in leptonic meson decays). The authors instead predict 10^-36 for this branching ratio. Are they accounting for neutrino masses? In that case, Dirac or Majorana? This should be clarified.
3. In Figure 3 (left), the labels of the various lines are put exactly on top of the 125GeV mass, where one would be more interested in reading the branching ratios of the SM Higgs from the plot. It is better if they are moved to another place.
4. In Eqs.3-4 the authors introduce the \delta_dir parameter, to describe the relative size of the direct contribution to the decay rate over the larger indirect one. Why then, in Table 2, they show the value of the value of -A_dir/A_ind instead of using \delta_dir, as is done in the other tables?
5. I believe that the values in the last column of Table 3 are wrong by one order of magnitude. 1/10 should be 1/100 (like it appears later, in Table 8), 1/20 > 1/200, etcetera.
6. In case of the decay H > gamma + (e+ e-)_1, an inconsistency of a factor 25 between the authors’s result and those from Ref. [79] is mentioned. What is the source of this rather large mismatch? Can the authors provide some more details about this?

Recommendation

Ask for minor revision

  • validity: -
  • significance: -
  • originality: -
  • clarity: -
  • formatting: -
  • grammar: -

Author:  Van Dung Le  on 2025-11-27  [id 6084]

(in reply to Report 2 on 2025-10-30)
Disclosure of Generative AI use

The comment author discloses that the following generative AI tools have been used in the preparation of this comment:

An AI tool was used to
1. Convert from LaTeX to Markdown,
2. Check for grammar
3. Assist in literature search
We used Gemini 3.0 and ChatGPT.

Warnings issued while processing user-supplied markup:

  • Inconsistency: Markdown and reStructuredText syntaxes are mixed. Markdown will be used.
    Add "#coerce:reST" or "#coerce:plain" as the first line of your text to force reStructuredText or no markup.
    You may also contact the helpdesk if the formatting is incorrect and you are unable to edit your text.

We thank the referee for the careful reading of our manuscript and the useful feedback provided. Here below, we provide an answer to all her/his questions as well as a verbatim list of updated text (if any) in the resubmitted version of our paper.

REFEREE: At the end of page 2 it it stated that precision tests of suppressed/forbidden processes in the SM have been mostly studied in B-decays so far. This is evidently not true, as even more powerful constraints on the scale of new physics are obtained from rare/forbidden Kaon decays, LFV tests in muon or tau decays, electric dipole moments, etc. The sentence should be modified.

ANSWER: Here, we meant to consider SM tests at multi-GeV masses and/or at colliders. To clarify this point better, the text has been updated as follows:

OLD:

Precision tests of suppressed or forbidden processes in the SM ---such as flavour changing neutral currents (FCNC), or processes violating lepton flavour (LFV) or lepton flavour universality (LFUV)--- are powerful probes of BSM physics that have been mostly studied in b-quark decays so far~\cite{LHCb:2018roe,Belle-II:2018jsg}.

NEW:

Precision tests of suppressed or forbidden processes in the SM ---such as flavour-changing neutral currents (FCNC), or processes violating lepton flavour (LFV) or lepton flavour universality (LFUV)--- are powerful probes of BSM physics. While highly competitive studies exist at low energies (e.g. in rare kaon decays, LFV searches with muons or taus, or electric dipole moment measurements), most investigations at multi-GeV mass scales have so far been mostly carried out exploiting b-quark decays at colliders~\cite{LHCb:2018roe,Belle-II:2018jsg}.

REFEREE: I don't understand why the SM prediction for $\rm H \to \nu \bar{\nu}$ is not exactly zero. In the SM, neutrinos are massless, and the decay rate of a scalar into two massless fermions of opposite helicity (as $\nu$ and $\bar{\nu}$ have) vanish due to angular momentum conservation (the rate is proportional to the fermion mass, as in leptonic meson decays). The authors instead predict $10^{-36}$ for this branching ratio. Are they accounting for neutrino masses? In that case, Dirac or Majorana? This should be clarified.

ANSWER: The $\rm H \to \nu\bar{\nu}$ decay is forbidden in the SM with massless neutrinos and the rate must be exactly zero, but neutrinos in nature are not massless (despite their mass generation mechanism being unknown), so the loop-induced decay considered here is in theory possible. Without any assumption on the coupling of the neutrinos to the SM Higgs boson, we estimated the size of this loop-induced decay which is heavily suppressed because of a chirality flip proportional to the (tiny) neutrino masses. Nonetheless, upon detailed re-investigation prompted by the referee's comment, we realized that our $\mathcal{O}(10^{-36})$ result originally derived with MG5@NLO was not correct. We have recalculated it and verified that it amounts to $\mathcal{O}(10^{-26})$ instead. In our resubmitted paper, we now explicitly state that our estimate assumes a nonzero neutrino mass, irrespective of their (unknown) mass generation mechanism (though we also provide now simple Yukawa-type estimates of this decay width assuming the neutrinos behave like all other Dirac fermions or as Majorana fermions) and perform the computation of the loop-induced process with their current upper mass value of $m_\nu \approx 0.1$ eV. This yields a numerically stable partial width with MG5@NLO, which can be treated as an upper limit for the loop-induced decay width considered here. To address all these concerns, the text has been updated as follows:

OLD:

Our first result is that of the invisible two-body Higgs boson decay into a neutrino pair $\rm H \to \nu\bar{\nu}$, which is infinitesimal in the SM ($\text{BR}\approx 10^{-36}$) compared to the standard invisible (four-neutrino) $\rm H \to ZZ^\star \to 4\nu$ decay ($\text{BR}\approx 0.1\%$)~\cite{Djouadi:2018xqq}. However, since the SM assumption of massless $\nu$'s is invalid, the $\rm H \to \nu\bar{\nu}$ decay can receive extra contributions depending on the mechanism of neutrino mass generation actually realized in nature.

NEW:

Our first result concerns the invisible two-body Higgs boson decay into a neutrino pair, $\rm H \to \nu\bar{\nu}$, proceeding via EW loops (Fig.~\ref{tab:H_decays_V_V_V} top left). In the SM with massless neutrinos, there is no tree-level Higgs-neutrino Yukawa coupling, and any amplitude for a scalar particle to decay into two massless fermions is forbidden by chirality (equivalently, a chirality flip is required and cannot be provided by massless fermions). However, in reality neutrinos are known to be massive, $m_\nu\lesssim 0.1$~eV, and therefore such an amplitude is allowed in principle, although it is proportional to the tiny neutrino mass (a chirality flip) and further suppressed by the weak coupling and loop factors. We estimate this pure loop-induced branching ratio with MG5@NLO to be of order $\text{BR}\approx 2 \times 10^{-26}$, i.e., utterly negligible compared with the dominant invisible four-neutrino decay, $\rm H \to ZZ^\star \to 4\nu$, which has a $\text{BR}\approx 0.1\%$, obtained from $\text{BR}(H\to ZZ^\star)\times\text{BR}(Z\to\nu\bar{\nu})^2$~\cite{Djouadi:2018xqq}.

For comparison, if neutrinos are Dirac fermions, a new right-handed neutrino field $\nu_R$ is added to the SM Lagrangian and they acquire mass via an ordinary Yukawa coupling ($y_\nu$) through $\mathcal{L}^\text{Yukawa} \sim y_\nu\bar L \tilde H\nu_R$, where $L$ is the lepton-handed lepton doublet and $\tilde H$ is the hypercharge-conjugated Higgs field. After EW symmetry breaking, $m_\nu=y_\nu v/\sqrt{2}$ with $v = 246$~GeV the Higgs vacuum expectation value, and the tree-level Higgs partial width into a neutrino pair can be derived with the usual fermionic formula,

$$ \Gamma(\mathrm{H}\to\nu\bar{\nu}) = \frac{G_\mathrm{F} m_\mathrm{H} m_\nu^2}{4\sqrt2\pi}\left(1-\frac{4m_\nu^2}{m_\mathrm{H}^2}\right)^{3/2}, $$
which for $m_\nu\lesssim 0.1$~eV yields $\Gamma(\mathrm{H}\to\nu\bar{\nu})\lesssim 8.2\times10^{-25}$~GeV, and therefore $\text{BR}(H\to\nu\bar{\nu})=\Gamma/\Gamma_\mathrm{H}^\mathrm{tot}\approx 2.0\times10^{-22}$, using $\Gamma_\mathrm{H}^\mathrm{tot} = 4.1\times10^{-3}$~GeV. Thus, a Dirac Yukawa-induced two-body decay (scaling as $m_\nu^2$) is still larger than the loop-induced branching ratio quoted above, but anyway utterly negligible for phenomenology. The result scales as $m_\nu^2$, so any smaller neutrino mass would further reduce the branching ratio. In an alternative case where neutrinos are Majorana fermions, and in the simplest assumption where the light Majorana masses arise from the Higgs mechanism so that the effective Higgs-neutrino coupling is still $y_\nu\propto m_\nu/v$, the partial width has the same $m_\nu^2$ scaling but is reduced by a 1/2 symmetry factor relative to the Dirac formula because the two final-state Majorana neutrinos are identical. Hence $\Gamma_\text{Majorana}(\mathrm{H}\to\nu\bar{\nu})\approx 1/2\Gamma_\text{Dirac}(H\to\nu\bar{\nu})$, and therefore $\Gamma(\mathrm{H}\to\nu\bar{\nu})\lesssim 4.1\times10^{-25}$~GeV and $\text{BR}(\mathrm{H}\to\nu\bar{\nu})\lesssim 1\times10^{-22}$. These estimates are valid for minimal Dirac- or Majorana-mass scenarios where the effective Higgs-neutrino coupling is proportional to $m_\nu/v$. Non-minimal BSM constructions (new light mediators, large neutrino-Higgs mixing, or exotic operators) could (by construction) enhance $\rm H \to \nu\bar{\nu}$, but such scenarios lie beyond the minimal SM-like Yukawa assumption and must be treated case by case.

OLD (Table 1 row):

$\rm H \, \to \; \nu+\overline{\nu} \quad 7.2\times 10^{-36}$ (this work)

NEW (Table 1 row):

$\rm H \, \to \; \nu+\overline{\nu} \quad 2.0\times 10^{-26}$ (this work)

REFEREE: In Figure 3 (left), the labels of the various lines are put exactly on top of the 125GeV mass, where one would be more interested in reading the branching ratios of the SM Higgs from the plot. It is better if they are moved to another place.

ANSWER: Fig. 3 left has been updated as suggested and, in addition, we have corrected the $\rm H \to \nu\bar{\nu}$ curve as per the discussion above. Also the right plot of Fig. 3 has been corrected so that the $\rm H \to \nu\bar{\nu}$ histogram matches the changes discussed above.

REFEREE: In Eqs. 3--4 the authors introduce the $\delta_{dir}$ parameter, to describe the relative size of the direct contribution to the decay rate over the larger indirect one. Why then, in Table 2, they show the value of the value of $-A_{dir}/A_{ind}$ instead of using $\delta_{dir}$, as is done in the other tables?

ANSWER: In Table 2, we list Higgs radiative decays to vector mesons ($\rm H \to \gamma+\text{VM}$) obtained with Eq. (2) that includes the direct and indirect amplitudes directly, whereas Tables 3 and 5 lists $\rm H \to Z,W\,+M$ decays which are described by Eqs. (3) and (4), where the $\delta_\mathrm{dir}$ is just derived as an approximate "correction" of the direct component. The $-A_{dir}/A_{ind}$ ratio provides an exact evaluation of the relative size of the direct and indirect amplitudes, whereas $\delta_\mathrm{dir}$ is just an indicative factor that gives the relative size of the direct over indirect branching fractions (and that is why we give only orders-of-magnitude for it in Tables 3 and 5). For the $\rm H \to Z,W\,+M$ decays, the $-A_{dir}/A_{ind}$ ratios are not straightforward to compute, as the interference pattern is complicated by imaginary parts arising from the transverse and longitudinal components of the VM (Eqs. 3--4), whereas their corresponding approximate $\delta_\mathrm{dir}$ ratio is easy to estimate.

REFEREE: I believe that the values in the last column of Table 3 are wrong by one order of magnitude. 1/10 should be 1/100 (like it appears later, in Table 8), 1/20 $\to$ 1/200, etc.

ANSWER: Thanks for catching this typo. Table 3 has been updated with the correct values:

OLD:

Table 3 last column: -- \; -- \; 1/10 \; -- \; -- \; 1/20 \; -- \; 1/10 \; 1/70 \; -- \; -- \; -- \; -- \;

NEW:

Table 3 last column: --\; -- \; 1/100 \; -- \; -- \; 1/200 \; -- \; 1/100 \; 1/700 \; -- \; -- \; -- \; -- \;

REFEREE: In case of the decay $\rm H \to \gamma + (e^+e^-)_1$, an inconsistency of a factor 25 between the authors's result and those from Ref. [79] is mentioned. What is the source of this rather large mismatch? Can the authors provide some more details about this?

ANSWER: We contacted the authors of Ref. [79] to understand the origin of the discrepancy. It comes from two sources. First, they include a more detailed W and quark loop-induced $\rm H\gamma\gamma$ coupling, while our work uses a point-like coupling derived from the $\rm H \to \gamma\gamma$ decay. Second, while the analytical expressions for the decay widths are correct, there had a mistake in their numerical evaluations of the partial widths (they used $m_e/2$ instead of $m_e$ mass in their expressions).

Their updated results are: $\mathcal{B}(H\to (e^+e^-)+\gamma) = 1.10\cdot10^{-11}$, $\mathcal{B}(H\to (\mu^+\mu^-)+\gamma = 1.12\cdot 10^{-11}$, $\mathcal{B}(H\to (\tau^+\tau^-)+\gamma = 3.48\cdot10^{-12}$, which are consistent with ours within a factor of three. The paper has been updated as follows:

OLD:

A recent work~\cite{Martynenko:2024rfj} has computed higher-order corrections to these decays, finding consistent results with ours except for the $\rm H \to \gamma + (e^+e^-)_1$ channel that would have about 25 times larger decay rates.

NEW:

A recent work~\cite{Martynenko:2024rfj} has computed higher-order corrections to these decays, finding consistent results with ours within a factor of three [footnote: Note that the value originally quoted in Ref.~\cite{Martynenko:2024rfj} had a typo that wrongly enhanced it by a factor of 25~\cite{MartinenkoPrivateComm}.]

OLD (Table X row):

$\rm H \to \gamma+(ee)_1 \quad (3.5-88) \times 10^{-12}$

NEW (Table X row):

$\rm H \to \gamma+(ee)_1 \quad (3.5-11) \times 10^{-12}$

---

## Round 2 · Author Response

Updated manuscript after feedback from referees.

---

## Round 2 · List of Changes

OLD: Precision tests of suppressed or forbidden processes in the SM ---such as flavour changing neutral currents (FCNC), or processes violating lepton flavour (LFV) or lepton flavour universality (LFUV)--- are powerful probes of BSM physics that have been mostly studied in b-quark decays so far~\cite{LHCb:2018roe,Belle-II:2018jsg}.
NEW: Precision tests of suppressed or forbidden processes in the SM ---such as flavour-changing neutral currents (FCNC), or processes violating lepton flavour (LFV) or lepton flavour universality (LFUV)--- are powerful probes of BSM physics. While highly competitive studies exist at low energies (e.g. in rare kaon decays, LFV searches with muons or taus, or electric dipole moment measurements), most investigations at multi-GeV mass scales have so far been mostly carried out exploiting b-quark decays at colliders~\cite{LHCb:2018roe,Belle-II:2018jsg}.

OLD: Our first result is that of the invisible two-body Higgs boson decay into a neutrino pair H→ν¯ν, which is infinitesimal in the SM (BR≈10−36) compared to the standard invisible (four-neutrino) H→ZZ⋆→4ν decay (BR≈0.1%)~\cite{Djouadi:2018xqq}. However, since the SM assumption of massless ν's is invalid, the H→ν¯ν decay can receive extra contributions depending on the mechanism of neutrino mass generation actually realized in nature.
NEW: Our first result concerns the invisible two-body Higgs boson decay into a neutrino pair, H→ν¯ν, proceeding via EW loops (Fig.~??? top left). In the SM with massless neutrinos, there is no tree-level Higgs-neutrino Yukawa coupling, and any amplitude for a scalar particle to decay into two massless fermions is forbidden by chirality (equivalently, a chirality flip is required and cannot be provided by massless fermions). However, in reality neutrinos are known to be massive, mν≲0.1~eV, and therefore such an amplitude is allowed in principle, although it is proportional to the tiny neutrino mass (a chirality flip) and further suppressed by the weak coupling and loop factors. We estimate this pure loop-induced branching ratio with MG5@NLO to be of order BR≈2×10−26, i.e., utterly negligible compared with the dominant invisible four-neutrino decay, H→ZZ⋆→4ν, which has a BR≈0.1%, obtained from BR(H→ZZ⋆)×BR(Z→ν¯ν)2~\cite{Djouadi:2018xqq}. For comparison, if neutrinos are Dirac fermions, a new right-handed neutrino field νR is added to the SM Lagrangian and they acquire mass via an ordinary Yukawa coupling (yν) through LYukawa∼yν¯L~HνR, where L is the lepton-handed lepton doublet and ~H is the hypercharge-conjugated Higgs field. After EW symmetry breaking, mν=yνv/√2 with v=246~GeV the Higgs vacuum expectation value, and the tree-level Higgs partial width into a neutrino pair can be derived with the usual fermionic formula, Γ(H→ν¯ν)=GFmHm2ν4√2π(1−4m2νm2H)3/2, which for mν≲0.1~eV yields Γ(H→ν¯ν)≲8.2×10−25~GeV, and therefore BR(H→ν¯ν)=Γ/ΓtotH≈2.0×10−22, using ΓtotH=4.1×10−3~GeV. Thus, a Dirac Yukawa-induced two-body decay (scaling as m2ν) is still larger than the loop-induced branching ratio quoted above, but anyway utterly negligible for phenomenology. The result scales as m2ν, so any smaller neutrino mass would further reduce the branching ratio. In an alternative case where neutrinos are Majorana fermions, and in the simplest assumption where the light Majorana masses arise from the Higgs mechanism so that the effective Higgs-neutrino coupling is still yν∝mν/v, the partial width has the same m2ν scaling but is reduced by a 1/2 symmetry factor relative to the Dirac formula because the two final-state Majorana neutrinos are identical. Hence ΓMajorana(H→ν¯ν)≈1/2ΓDirac(H→ν¯ν), and therefore Γ(H→ν¯ν)≲4.1×10−25~GeV and BR(H→ν¯ν)≲1×10−22. These estimates are valid for minimal Dirac- or Majorana-mass scenarios where the effective Higgs-neutrino coupling is proportional to mν/v. Non-minimal BSM constructions (new light mediators, large neutrino-Higgs mixing, or exotic operators) could (by construction) enhance H→ν¯ν, but such scenarios lie beyond the minimal SM-like Yukawa assumption and must be treated case by case.

OLD (Table 1 row):H→ν+¯¯¯ν7.2×10−36 (this work)
NEW (Table 1 row): H→ν+¯¯¯ν2.0×10−26(this work)

Fig. 3 left has been updated as suggested and, in addition, we have corrected the H→ν¯ν curve as per the discussion above. Also the right plot of Fig. 3 has been corrected so that the H→ν¯ν histogram matches the changes discussed above.

OLD: Table 3 last column: -- \; -- \; 1/10 \; -- \; -- \; 1/20 \; -- \; 1/10 \; 1/70 \; -- \; -- \; -- \; -- \;
NEW: Table 3 last column: --\; -- \; 1/100 \; -- \; -- \; 1/200 \; -- \; 1/100 \; 1/700 \; -- \; -- \; -- \; -- \;

OLD: A recent work~\cite{Martynenko:2024rfj} has computed higher-order corrections to these decays, finding consistent results with ours except for the H→γ+(e+e−)_1 channel that would have about 25 times larger decay rates.
NEW: A recent work~\cite{Martynenko:2024rfj} has computed higher-order corrections to these decays, finding consistent results with ours within a factor of three [footnote: Note that the value originally quoted in Ref.~\cite{Martynenko:2024rfj} had a typo that wrongly enhanced it by a factor of 25~\cite{MartinenkoPrivateComm}.]

OLD (Table 6 row): H→γ+(ee)_1 (3.5−88)×10−12
NEW (Table 6 row): H→γ+(ee)_1 (3.5−11)×10−12

OLD: with MG5@NLO using the SMEFT@NLO model~\cite{Degrande:2020evl}.
NEW: with MG5@NLO using the encoded loop-induced SM effective couplings of the SMEFT@NLO model~\cite{Degrande:2020evl}.

---

## Editorial Decision

refereeing_in_preparation